# Representation Shattering in Transformers:
# A Synthetic Study with Knowledge Editing

**Kento Nishi** [1 2 3]   **Rahul Ramesh** [4]   **Maya Okawa** [2 3]   **Mikail Khona** [5]
**Hidenori Tanaka**[* 2 3]   **Ekdeep Singh Lubana**[* 2 3]

## Abstract

Knowledge Editing (KE) algorithms alter models' weights to perform targeted updates to incorrect, outdated, or otherwise unwanted factual associations. However, recent work has shown that applying KE can adversely affect models' broader factual recall accuracy and diminish their reasoning abilities. Although these studies give insights into the potential harms of KE algorithms, e.g., performance evaluations on benchmarks, little is understood about *why* such destructive failures occur. Motivated by this, we define a novel synthetic task in which a Transformer is trained from scratch to internalize a "structured" knowledge graph. The structure enforces relationships between entities of the graph, such that editing a factual association has "trickling effects" on other entities (e.g., altering X's parent is Y to Z affects who X's siblings' parent is). Through evaluations of edited models on this task, we show that KE inadvertently affects representations of entities beyond the targeted one, distorting relevant structures that allow a model to infer unseen knowledge about an entity. We call this phenomenon *representation shattering* and demonstrate that it degrades models' factual recall and reasoning performance. We further corroborate our findings in naturalistic settings with pre-trained Llama and Mamba models as well. Overall, our work yields a precise mechanistic hypothesis to explain why KE has adverse effects on model abilities.

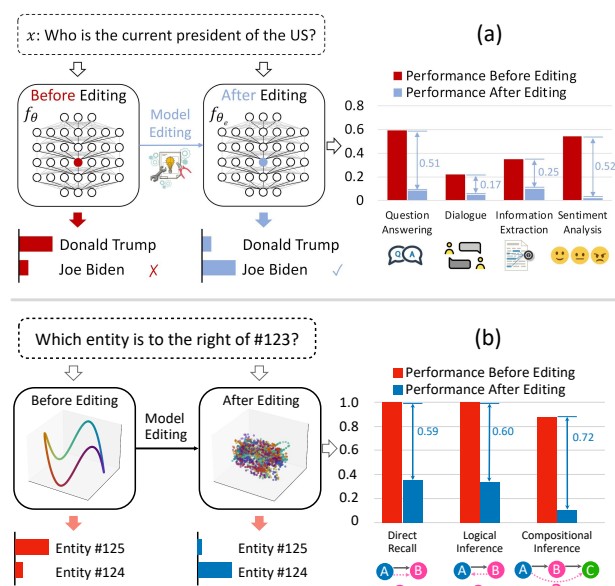

*Figure 1.* **Representation shattering** **as a mechanistic hypothesis for why knowledge editing has adverse effects on models' general capabilities.** *(a)* Prior works find that editing facts, e.g., the president of the US, can harm general abilities of LLMs (figure reproduced from Gu et al. (2024)). *(b)* We introduce a synthetic data generation process (DGP) defined by a knowledge graph containing ring-shaped geometries. Training a model on this data, we observe that the model's internal representations mirror the ring structure of the underlying DGP. We explain the post-edit degradation of the model's capabilities by uncovering the "shattering" of latent representations. E.g., in the provided illustration, while the edit successfully changes the fact (Entity #124 is to the right of Entity #123), it also degrades model's broader performance.

---
[*]Equal contribution [1]Harvard College [2]CBS-NTT Program in Physics of Intelligence, Harvard University [3]Physics and Informatics Lab, NTT Research Inc. [4]Computer and Information Science, University of Pennsylvania [5]Department of Physics, Massachusetts Institute of Technology. Correspondence to: Kento Nishi <kentonishi@college.harvard.edu>, Ekdeep Singh Lubana <ekdeeplubana@fas.harvard.edu>, Hidenori Tanaka <hidenori_tanaka@fas.harvard.edu>.

*Proceedings of the 42nd International Conference on Machine Learning*, Vancouver, Canada. PMLR 267, 2025. Copyright 2025 by the author(s).

## 1. Introduction

Large language models (LLMs) have led to unprecedented advances in several domains (Gemini Team, 2023; Bubeck et al., 2023; Touvron et al., 2023; Thoppilan et al., 2022; Chowdhery et al., 2022; Qin et al., 2023; Chen et al., 2021; Ahn et al., 2022; Driess et al., 2023). However, the static nature of their training pipelines implies that as our world evolves, models' internalized knowledge can become incorrect or outdated. To address this, recent work has proposed several protocols for knowledge editing (KE), wherein the

goal is to minimally and precisely alter model weights such that only the targeted information (and its relevant associations) are updated, but all unrelated information remains (ideally) unaffected (Mitchell et al., 2022; Meng et al., 2022a; 2023; Dai et al., 2021; Cheng et al., 2023; De Cao et al., 2021; Sinitsin et al., 2020).

Despite significant work on the topic, it still remains unclear precisely what effects KE should have on a model. For example, assume you edit the fact that "*Michael Jordan* won the 1998 NBA MVP" to "*Reggie Miller* won the 1998 NBA MVP", then what should the impact of such an edit be? Should the model now believe Michael Jordan and the Chicago Bulls never reached the NBA finals in 1998? Should it perhaps believe Reggie Miller was on the Chicago Bulls? Should the pop quote "Be like Mike" (Wikipedia, 2024) now become "Be like Reggie"? As Hofweber et al. (2024); Hase et al. (2024) argue, it is difficult to design clear, well-defined answers for such questions. Motivated by this, recent work has started investigating precisely what effects KE actually has on the model (Hoelscher-Obermaier et al., 2023; Li et al., 2023b; Lynch et al., 2024). For example, Cohen et al. (2023) demonstrate that knowledge beyond the edited fact can often be impacted such that the model begins to have an incoherent understanding of the world; Gupta et al. (2024a) demonstrate unrelated facts are often forgotten by the model post-editing; and Gu et al. (2024) show that KE can harm broader reasoning capabilities beyond mere factual recall. While these works clearly demonstrate the detrimental impacts of editing on a model, they still leave open the question precisely *why* such harms occur—at a mechanistic level, how are model representations impacted such that a broad set of knowledge and capabilities in a model are heavily distorted once an edit occurs?

**This work.** To address the questions above, we aim to develop a mechanistic understanding of the impact of KE on a model's internals. For this purpose, we argue we must solve two problems: (i) identify how a model expresses knowledge about some predefined set of entities in its representations, and (ii) investigate how this mechanism is affected as we apply KE to alter a fact corresponding to a subset of the entities. Instead of attacking a complicated system that may be difficult to interpret (e.g., an off-the-shelf LLM), we take inspiration from a multitude of recent papers that establish synthetic abstractions of the target system and develop precise hypotheses as to why the phenomenon-in-question occurs (Allen-Zhu & Li, 2023c;a;b; Okawa et al., 2023; Chan et al., 2022; Li et al., 2023a; Lubana et al., 2024). Specifically, we define a data-generating process that yields entities arranged in a *structured* knowledge graph. This structure is defined via use of a predefined set of relations that *locally* constrain how entities relate to each other (similar to parent-child relations). Given enough entities and relations, such local constraints manifest a broader *global* structure in the

knowledge graph. Performing traversal over the nodes of this knowledge graph, we get sequences that can be used as "strings" to train a Transformer on. As we show, this protocol leads to the model precisely encoding the structure of the graph in its latent representations. However, when KE is applied to edit either incorrectly learned facts or insert counterfactual knowledge (using the method proposed by Meng et al. (2022a)), we find latent representations are heavily distorted and the graph structure completely destroyed—we call this phenomenon **representation shattering**. Interestingly, this phenomenon manifests in proportion to how far the proposed edit moves a given node from its current location to a new location in the graph (defined via edge distance). We thus hypothesize representation shattering underlies the detrimental effects of KE on a pretrained model's factual and reasoning capabilities at broad. Overall, we make the following contributions in this work.

- **Structured Knowledge Graphs as a Toy Setting for Investigating Impact of KE.** We propose use of a structured knowledge graph wherein entities (nodes) are connected to each other via predefined local constraints (relations) that manifest into a broader, global structure in the graph (see Sec. 3). Training Transformers on strings (path traversals) from the graph, we find model representations precisely encode the global structure of the graph. This allows us to assess the impact of KE at a more mechanistic level, since distorting a fact now has global effects that can be precisely (and, in fact, visually) delineated by analyzing the model representations.

- **Representation Shattering as a Mechanistic Hypothesis to Explain Detrimental Effects of KE.** We find KE distorts latent representations for entities in the graph such that the global geometry learned during pretraining is, at times, completely destroyed—we call this phenomenon **representation shattering** and hypothesize it underlies the detrimental effects of KE on model capabilities observed in prior work (see Sec. 4 and Fig. 1). As we show, the extent of harm on latent representations turns out to be correlated to the amount an edit alters the graph from its original organization into the new, desired one.

- **Investigations with Off-the-Shelf LLMs.** Using pretrained Llama and Mamba models, we provide evidence for our claims about representation shattering in more naturalistic settings. For one, we find real-world analogues to our synthetic knowledge graph structures (i.e., days of the week) yield similar shattering phenomena in Llama and Mamba models to what we observe in our toy setup (see Sec. 4.5). Additionally, we further reinforce the generality of our findings with preliminary replications of representation shattering under more complex knowledge graph geometries, such as trees (i.e. countries and their cities, Appx. H.3).

## 2. Related Work

**Knowledge Editing.** Several protocols for knowledge editing (KE) have been proposed in recent work. Early work defined meta-learning based approaches (Sinitsin et al., 2020; De Cao et al., 2021; Mitchell et al., 2022) and established the broader desiderata for what properties a KE protocol should satisfy; e.g., ensuring facts unrelated to the target one are not hampered via the editing protocol. Building on work aimed at understanding how Transformers encode knowledge in their internals (Geva et al., 2020), modern KE protocols focus on defining closed-form operations that involve (i) localizing knowledge to specific components in a model (e.g., MLP layers) and (ii) defining operations to alter a factual association by assuming the fact is represented in a localized manner (Meng et al., 2022a; 2023).

**Evaluations of Knowledge Editing Methods.** As argued by Hase et al. (2024); Hofweber et al. (2024), the problem of KE is relatively ill-defined. Consequently, it is unclear that when we edit knowledge within a model, what effects said edits *should have* on other facts it may have internalized during training. Prior work has hence taken an alternative approach, primarily focusing on developing an empirical understanding of what the phenomenology of KE protocols is: e.g., if an edit is performed, how are counterfactual statements or unrelated facts affected. These works generally show that KE in fact has extreme detrimental effects on a model, e.g., hampering both its broader internalized knowledge and its reasoning abilities (Hase et al., 2023; Cohen et al., 2023; Hoelscher-Obermaier et al., 2023; Gupta et al., 2024a; Gu et al., 2024). While the primary methodology used in such papers is to perform empirical benchmarking of a model that has undergone editing, we instead focus on a mechanistic analysis of how editing alters a model's representations (albeit primarily in a toy synthetic task) to yield the undesirable effects on model abilities.

**Explaining Models via Synthetic Tasks.** To disentangle the failures of KE methods from the failures of the models themselves, we argue for use of a more controllable and interpretable setup. Such a setup can help identify a concrete hypothesis for why KE has undesirable effects on the model, which we can then analyze in naturalistic settings by designing more precisely defined experiments. This methodology of designing toy, control tasks to investigate hypotheses for phenomenology of a neural network has yielded promising results in recent years, providing, e.g., a concrete hypothesis for how chain-of-thought reasoning aids model capabilities (Prystawski et al., 2024; Feng et al., 2023), models for emergent capabilities (Okawa et al., 2023; Lubana et al., 2024), existence of multidimensional representations (Engels et al., 2024), and failure modes for compositional generalization (Zhou et al., 2023).

## 3. Formalzing Knowledge Editing

Epistemology has grappled with the nature of knowledge for centuries (Chappell, 2005). In this work, we adopt a humble yet precise definition of knowledge based on structured knowledge graphs. A knowledge graph is used to represent how facts, entities, and relations are interlinked, giving rise to notions of consistency, coherency, and reasoning across different pieces of information. Using these definitions, we will define a synthetic data generation process on knowledge graphs that enables a systematic study of knowledge editing in Transformers.

### 3.1. Knowledge Graphs

A knowledge graph consists of a collection of entities $X = \{x_i\}_{i=1}^n$, and a collection of facts $F$ that relate different entities. For example, a graph defined on entities $X = \{$"Alice", "Bob", "Carol"$\}$ can encode the fact "Alice is the advisor of Bob" using the relation "advisor", represented as ("Alice", "advisor", "Bob").

**Definition 3.1** (**Knowledge graph**). Formally, a knowledge graph $G = (X, R, F)$ consists of nodes $X$, relations $R$, and facts $F$, where each fact $f = (x_i, r, x_j) \in F$ is defined by a relation $r \in R$ between two entities $x_i, x_j \in X$.

A **relation sub-graph** corresponds to a sub-graph constructed by only considering facts that use relation $r$. For example, $G_{\text{advisor}}$ is a sub-graph that specifies all facts for the relation "advisor". Every knowledge graph contains a collection of facts that can be inferred from the graph.

Related pieces of information such as "Alice's advisor was Bob" and "Bob's advisor was Carol" can be composed to form cohesive statements such as "Alice's advisor's advisor was Carol. To capture such statements, we define compositions of relations. The composition of relations are essential to capture ripple effects that occur in the knowledge graph after an edit (Cohen et al., 2023) to a relation in $R$.

**Definition 3.2** (**Composition of relations**). A composition of relations $\vec{r} = (r_1, r_2, \cdots, r_k) \in R^k$ with respect to knowledge graph $G$ is defined such that for every fact $f = (x_i, \vec{r}, x_j)$, there exists a collection of facts $\{(x_i, r_i, x_{i+1})\}_{i=1}^k$ for which $x_1 = x_i$ and $x_{k+1} = x_j$. In other words, any fact defined on the composition of relations has a corresponding set of facts defined on relations from $R$. Furthermore, the set of facts form a path in the knowledge graph such that the sequence of relations in the path are $r_1, r_2, \cdots r_k$.

### 3.2. Cyclic Graphs: Description of Entities/Relations

We mainly study knowledge graphs where every relation sub-graph is a set of disjoint cyclic graphs, i.e., for every entity $x_i$ and relation $r$, there exists exactly one entity $x_j$ such that $(x_i, r, x_j) \in F$. We specifically choose a cyclic

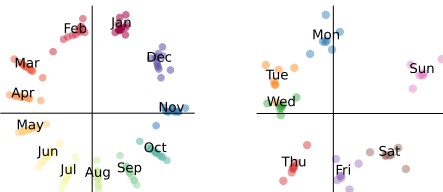

*Figure 2.* **Isomap projections of representations in Llama-3.1-405B (Fiotto-Kaufman et al., 2024).** The geometry of the data—for example, the cyclic nature of months or days—is often reflected in the representations learned by language models. Similar representations can also be found in other models like GPT-2-Small and Mistral 7B (Engels et al., 2024).

geometry as a global constraint on the graph structure since cycles are a common pattern that relate entities in natural language domains; e.g., see Fig. 2, where we show a 2D projection of representations from Llama-3.1-405B corresponding to months of a year and days of the week naturally organize in a cyclic fashion. For supplementary explorations of geometries of concepts other than cycles (i.e. trees), refer to Appx. H.3.

Knowledge editing methods, e.g., ROME (Meng et al., 2022a) and MEMIT (Meng et al., 2023), target a set of entities for which predefined facts are to be edited, while using another retain set of facts about said entities to help ensure relations beyond the targeted ones are not altered. A test set of facts are then used to evaluate how well the method worked. Motivated by this, we define a knowledge graph with 2048 entities (denoted by 1-2048) over which we define 3 cyclic orders (order I, II and III). The cyclic orders are generated using random permutations of the entities. We create 8 relations for each cyclic order totaling to 24 relations. The 8 relations correspond to the 1-hop, 2-hop, 3-hop and 4-hop neighbors in the clockwise and anti-clockwise directions in the cycle. The relations are named after a combination of the cyclic order (I, II, III), the neighbor's distance (between 1-4) and the neighbor's direction (Clockwise, Anti-clockwise). For instance, the relation "I_C2" denotes the 2-hop neighbor in the clockwise direction, with respect to cyclic order I. The 1-hop neighbor relation graphs (both clockwise and anti-clockwise) contain a single cycle, 2-hop relation graphs consist of 2 cycles, the 3-hop relation graph contains 1 cycle, while the 4-hop relation graph contains 4 cycles. The k-hop neighbor relations are related to each other by design, so any edit to one k-hop relation should be consistent with all other k-hop relations. An edit corresponds to changing a fact in the knowledge graph and can also be interpreted as changing an edge in the relation graph. For an illustrative example, see Fig. 3.

Depending on the fact being edited, the 3 cyclic orders are used to define the edit sub-graph, the retain sub-graph, and the test sub-graph. We need 3 cyclic orders because knowledge editing methods target **edit sub-graph relations**; the

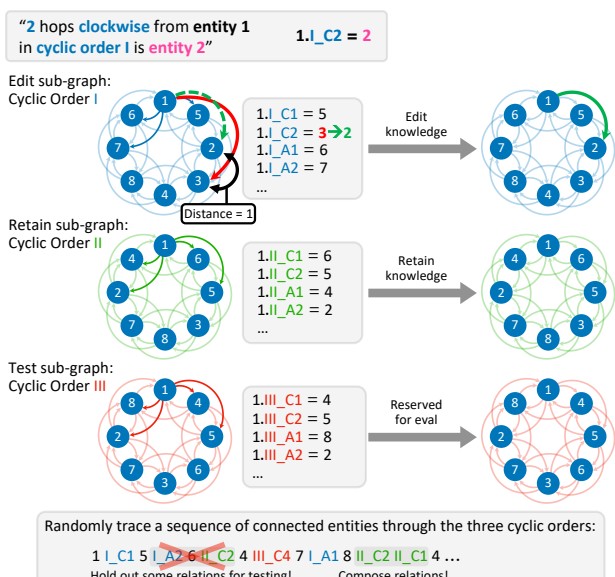

*Figure 3.* **Synthetic data generation process with a cyclic knowledge graph.** The entities (nodes) are arranged according to 3 different cyclic orders. Each entity (node) has relations (directed edges) pointing to 8 other entities in each cyclic order which totals to 24 relations across all 3 orders. The relations correspond to 1-4 hop neighbors in the clockwise and anti-clockwise directions. We select a random path on the knowledge graph using all 24 relations to generate a prompt (shown above). The relations follow the naming convention of ⟨cyclic-order⟩_⟨direction⟩⟨hops⟩, i.e. II_A3 is the relation corresponding to the three-hop anti-clockwise neighbor in the second cyclic order. In cyclic order I, the above figure denotes an edit for a relation between Entity 1 to Entity 3 (red) to a relation between Entity 1 to Entity 2 (green). The distance of the edit is 1, as defined with respect to cyclic order I.

facts based on these edit relations are then tested to check if a knowledge edit was successful. Meanwhile, the **retain sub-graph relations** are used by the knowledge editing algorithm to minimize changes to unrelated relations, but no edits are made to facts that use these relations. Finally, the **test sub-graph relations** are used to define facts that are neither directly edited, nor used by the knowledge editing algorithm. The relations are used to evaluate whether unrelated facts remain unchanged after a knowledge edit. We note that relations for all 3 sub-graphs are seen during pre-training and this distinction between the cyclic orders is made *only* during model editing. Finally, the **distance** of an edit (shown in Fig. 3) is defined as the shortest distance between the original and edited entity in the cyclic order.

### 3.3. Experimental Setup

**Data generation process.** We generate a sequence of alternating entities and relations resembling $x_1 \vec{r}_1 x_2 \vec{r}_2 x_3 \vec{r}_3 \ldots$, where any consecutive triplet of entity, relation, and entity $x_i r_i x_{i+1}$ from the sequence is a fact $(x_i, \vec{r}_i, x_{i+1})$ in the knowledge graph. The composition of relations

$\vec{r}_i = r_{i1} r_{i2} r_{i3} \dots$ is a sequence of 1 or more relation tokens, while $x_i$ is a single entity token. Every token is sampled using a uniform probability over all the permissible choices (see Alg. 1). For example, a plausible sequence for the example in Fig. 3 is "1 I_C4 4 III_A2 8 III_A3 3 II_C2 7", which is an alternating sequence of entities and k-hop relations. As previously noted, relations belonging to all three cyclic orders are included in the data generation process; the distinction between edit, retain, and test relations is only relevant to knowledge editing on a trained model. Furthermore, we remark that this sampling process is identical to traversing random walks on the knowledge graph, similar to previous works (Prystawski et al., 2024; Khona et al., 2024). Additional details of the generation process are documented in Appx. A and Appx. B.

**Training setup.** We train a Transformer model using next-token prediction on the synthetic data generated from the above data generation process. For all experiments (unless stated otherwise), we use a 2-layer nanoGPT Transformer (Karpathy, 2021). For additional details about the toy model, see Appx. D. For experiments with real-world LLMs, see Sec. 4.5.

**Evaluation (seen facts).** We assess the model's ability to remember facts seen during training, both before and after an edit. Specifically, to check if the model has learned the fact $(x_i, \vec{r}, x_j)$, we prompt it with an entity $x_i$ and a relation $\vec{r}$, expecting it to produce $x_j$ as the next token. In practice, the model outputs can vary across prompts: we account for this by averaging the softmax probabilities across 5 randomly sampled sequences of the form `ctx` $x_i \vec{r}$ and using the output token with the highest probability. Herein, `ctx` $x_i$ denotes a randomly sampled sequence that ends in $x_i$.

**Evaluation (unseen facts).** We also evaluate the model on two other criteria. *(1)* **Compositional inference.** In addition to facts seen in the training data, we evaluate the model on compositions of relations. The model must preserve geometric structures of the data in order to compositionally generalize after a knowledge edit. *(2)* **Logical inference.** A key feature of reasoning in natural language is logical inference. For example, if Alice is said to be the advisor of Bob, then Bob is an advisee of Alice (even if it is not explicitly stated). Our data generation process has similar relations, such as clockwise and anti-clockwise 1-hop neighbors. By "holding out" one direction for some such pairs of relations from being observed verbatim in the training dataset, i.e., the relation may only appear compositionally, we can assess the degree to which the model internalizes properties among related relations. We can also evaluate if editing a fact for a relation changes the fact for other related relations, i.e., we check if the model's knowledge is logically self-consistent after an edit. Precise prompt formats for both (1) and (2) are documented in Appx. C.

### 3.4. Representation Shattering

In this work, we explore the hypothesis that knowledge editing methods distort the geometry of the representations of entities in the knowledge graph. We argue that this distortion can give us insight into why knowledge editing degrades the general capabilities of a model. More precisely, in the following sections, we investigate the following hypothesis.

*Hypothesis* 3.3 (**Representation shattering**). *Language models embed related entities on a manifold in their internal representations. KE methods distort this manifold in order to insert new facts or alter old ones, i.e., they shatter model representations. The extent of representation shattering increases with the distance between the old fact and the desired new fact on the manifold.*

To quantify the extent of representation shattering, we define a precise metric to capture the amount of distortion of the representations:

$$R(D_*) = \frac{||D_* - D_\varnothing||_F}{||D_\varnothing||_F}, \tag{1}$$

where $||D||_F$ is the Frobenius norm of $D$, $D_\varnothing$ is the pairwise distance matrix of the entities computed using the unedited model, and $D_*$ is the pairwise distance matrix computed using the edited model. $D_*$ and $D_\varnothing$ are both $n \times n$ matrices over $n$ entities, and entry $D_{ij}$ is the Euclidean distance between the representation vectors of $x_i$ and $x_j$. The matrices are computed over only entity tokens (relation tokens are excluded).

Note that $R(D_*)$ is permutation-sensitive by design: a zero value indicates that every entity token preserves its location in representation space. Isomorphic geometries (e.g., wherein representations of two entity tokens swap positions in the manifold) still yield nonzero $R(D_*)$ values, because the relative positions of entities (as identified by their respective pre-assigned tokens) must have been distorted to achieve the edit(s).

## 4. Uncovering Representation Shattering

We study knowledge editing methods like ROME (Meng et al., 2022a), MEMIT (Meng et al., 2022b), PMET (Li et al., 2024), and AlphaEdit (Fang et al., 2024) in this work. While in the main paper we primarily present results with ROME (see Appx. E.1 for a short primer), we provide results with other methods in Appx. G.1.1 and Appx. G.1.2. We also verify the assumptions made by these methods in Appx. G.2.

We perform two different types of edits: corrective edits and counterfactual edits. **Corrective edits** are applied to facts which the model recalls *incorrectly* after training. A **counterfactual edit** introduces a new fact, i.e., it changes fact $(x_i, r, x_j)$ to fact $(x_i, r, x_k)$ where $x_j \neq x_k$. Such an edit introduces inconsistencies in the knowledge graph.

| | | (a) Unedited | (b) Corrective | | (c) Counterfactual | | | |
|---|---|---|---|---|---|---|---|---|
| | Cyclic Order | Acc. | Sub-Graph | ⟨ΔAcc.⟩ | Distance-$d$ Edit ⟨ΔAcc.⟩ | | | |
| | | | | | $d$=1 | $d$=2 | $d$=3 | $d$=4 |
| Direct Recall | I | 98.34 | Edit | -21.95 | -01.49 | -67.01 | -77.07 | -77.94 |
| | II | 93.71 | Retain | -22.64 | -01.91 | -66.70 | -75.49 | -75.42 |
| | III | 99.37 | Test | -21.83 | -01.75 | -67.00 | -76.12 | -77.90 |
| Logi. Infer. | I | 98.16 | Edit | -22.24 | -01.44 | -67.22 | -77.14 | -78.02 |
| | II | 93.95 | Retain | -22.50 | -01.83 | -66.88 | -75.67 | -75.67 |
| | III | 99.40 | Test | -22.03 | -01.80 | -67.31 | -76.27 | -78.23 |
| Comp. Infer. | I | 88.15 | Edit | -29.60 | -05.32 | -73.15 | -80.35 | -80.63 |
| | II | 79.31 | Retain | -31.92 | -05.32 | -71.21 | -78.70 | -78.87 |
| | III | 93.50 | Test | -31.70 | -06.69 | -74.88 | -81.38 | -80.62 |

*Table 1.* **The direct recall, logical inference, and compositional inference accuracies before and after KE.** Results are for ROME; see Appx. G.1.1 for other methods. *(a)* The performance of our model (before editing) across the three cyclic orders (I, II, and III). Not only does our model perform well on direct recall, but it also generalizes to both logical and compositional inference tasks. This suggests that the model's internal representations extend beyond simple memorization and capture the underlying global structure that relates entities. *(b)* Changes in model accuracy after applying corrective knowledge edits. Each ⟨ΔAcc.⟩ result is averaged across multiple edits, and each row labeled edit/retain/test is averaged across each of the cyclic orders *taking turns*, i.e., playing the roles of the edit, retain, and test sub-graphs. We find that corrective knowledge edits negatively affect the model's accuracy both on related and unrelated facts. These results align with the findings on LLMs (Gu et al., 2024; Gupta et al., 2024a). *(c)* ⟨ΔAcc.⟩ for edit, retain, and test sub-graphs after applying counterfactual edits. Intentionally introducing inconsistencies into the model's knowledge via counterfactual KE can significantly degrade model capabilities. Furthermore, the greater the induced inconsistency (scaling the counterfactual edit distance $d$ from 1-4), the more severe the resulting performance degradation.

Overall, we show the following. *(1)* Transformers trained on knowledge graphs recall facts, perform logical inferences, and compositional inferences. However, *both corrective and counterfactual edits degrade the model on all three fronts* (Sec. 4.1). *(2)* Transformers learn a representation that reflects the underlying geometry of the data. *Knowledge edits "shatter" this representation, which serves as an explanation for the degradation in accuracy after KE* (Sec. 4.2). *(3)* Counterfactual edits with larger distance display a larger degree of shattering (Sec. 4.4). *(4) These phenomena occur in pretrained language models*, indicating representation shattering can explain the degradation seen in model abilities after KE (Sec. 4.5).

### 4.1. Knowledge Editing can Degrade Model Accuracy

We evaluate the effects of counterfactual and corrective edits on three fronts. **Direct recall accuracy** calculates the accuracy of facts seen during training. **Logical inference accuracy** measures the accuracy on a subset of held out relations that can be inferred from other relations, i.e., the k-

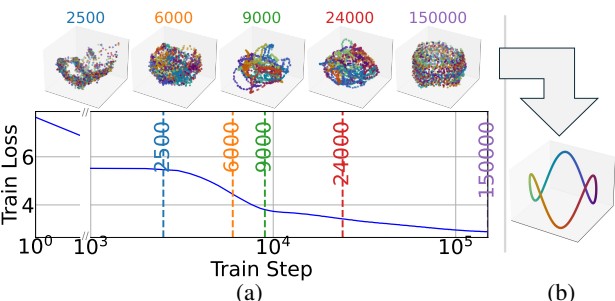

(a)            (b)

*Figure 4.* **Transformers learn representations that mirror the geometry of the underlying data.** *(a)* The representations—or output of the second attention layer—for the input $x\vec{r}$ for different entities $x$ and fixed relation $\vec{r}$ are visualized using Isomap. The model learns the cyclic ordering to represent all the facts. For visualizations of representations at various other model components and relation prompts, please see Appx. G.4 and Appx. G.5. *(b)* To improve the visual fidelity of the projected representations when comparing post-edit models to the unedited model, we pre-process the Isomap neighborhood graphs using the outputs of the Transformer model. For more details, see Appx. F.

hop anti-clockwise neighbors can be inferred directly from the k-hop clockwise neighbors. **Compositional inference accuracy** measures the accuracy on a held out subset of compositions of two relations. Both logical inference and compositional inference measure the accuracy on samples that would be considered *out-of-distribution*.

We report scores for all three metrics in Tab. 1. The model's logical and compositional inference accuracies are close to the direct recall accuracy, which implies that the model generalizes outside of the training data before KE. **However, after KE, all accuracies decrease, with a more severe decrease for counterfactual edits** (they introduce inconsistencies between facts).

### 4.2. Model Representations Capture Data Geometry

The model achieves high compositional and logical inference accuracies before knowledge editing, indicating that it captures the global structure of the data and does not merely memorize all the facts seen during training. We see this reflected in the internal representation of the model (output of the second attention layer), which we visualize using Isomap (Tenenbaum et al., 2000). Isomap preserves geodesic distances along the manifold, which is essential for visualizing non-linear geometries—e.g., rings and torii (Chaudhuri et al., 2019; Khona & Fiete, 2022). Meanwhile, PCA, being linear, can collapse or stretch such structures. This makes us prefer Isomap for our analysis, since faithful projections of the original topology turn out to be critical for interpreting how KE affects global geometry. We use PCA as well when linearity suffices (see Appx. G.2.2 and Appx. G.3).

In Fig. 4a, we plot how the Isomap embeddings of the inter-

| Sub-Graph | $d=1$ | $d=2$ | $d=3$ | $d=4$ |
|---|---|---|---|---|
| Edit | 01.80 | 21.93 | 26.22 | 27.90 |
| Retain | 01.80 | 20.84 | 25.32 | 27.28 |
| Test | 01.84 | 21.89 | 26.52 | 28.68 |

*Table 2.* Mean $R(D_*)$ for counterfactual edits, averaged across each sub-graph type. We observe higher degrees of representation shattering for greater counterfactual edit distances ($d$). Results are for ROME; other methods also reproduce this relationship (Appx. G.1.2).

nal representations for the input with one entity and relation ($x\vec{r}$) evolves over the course of training. The different data points correspond to different values of the entity $x$, for a fixed relation $\vec{r}$ and the points in the plot are colored by the cyclic ordering. **We see that the representation manifold resembles the cyclic ordering of the entities, particularly towards the end of training.** In addition, we confirm that the model correctly internalizes the independence of the cyclic orders I, II, and III in its representations (see Appx. G.3 for a visualization).

### 4.3. Corrective KE Shatters Representation Geometry

We assess how the representation changes after applying a corrective knowledge edit—i.e., applying KE to a fact that the model learned incorrectly during training. While one would expect the performance of the model to increase after a corrective edit, we find the opposite: *a corrective edit results in a drop in all accuracies* (see Tab. 1). These results align with previous empirical findings showing that reasoning capabilities degrade after corrective edits (Gu et al., 2024; Cohen et al., 2023).

We visualize the representations of 3 different models using the techniques described in Sec. 4.2 and Fig. 4b. The 3 models are obtained after applying 3 different edits and are selected to have high (★), intermediate (▲), and low (✖) direct recall accuracies. In Fig. 5, we observe that the model with the highest accuracy (★) has a representation that preserves the geometry of the data after the edit. However, as the model accuracy decreases, the representations also display a greater degree of distortion, no longer capturing the geometry of the data; in other words, the model is affected by representation shattering. Beyond visual inspection, this trend is also quantified in Fig. 5c, which shows a strong negative relationship between the distortion metric $R(D_*)$ (Eq. 1) and model accuracy ($r^2 = 0.905$).

### 4.4. Shattering Scales with Counterfactual Edit Distance

Counterfactual editing, wherein ones adds new facts that were unseen during training, is commonly used for evaluating KE protocols (Meng et al., 2022a; 2023; Gupta et al., 2024a; Hoelscher-Obermaier et al., 2023). We consider 25 different counterfactual edits corresponding to every single

counterfactual edit distance, where the counterfactual edit distance (or CE distance) is the distance between the entity in the old fact and new fact as measured in the cyclic order. Fig. 3 illustrates an example where the counterfactual edit has an edit distance of 1. In Fig. 6, **we see that increasing the distance of the counterfactual edit results in a drop in accuracy and an increasing in the extent of shattering**. This relationship is numerically supported by $R(D_*)$, as shown in Tab. 2: shattering increases as counterfactual edit distance increases. In other words, when a new fact changes one entity to another, the extent of shattering increases as the distance between the old and new entity increases. As a naturalistic parallel, if the entities are different months, accuracy is higher when we edit "December" to "November" as opposed to "July".

Summarizing these experiments, we find that for a given subject entity, edits with larger CE distances imply greater displacement in the representation manifold and higher $R(D_*)$—in reverse, higher $R(D_*)$ implies a larger CE distance. This manipulation approximates an intervention on representation geometry itself, and this offers compelling evidence for a causal link between shattering and performance loss. This is consistent with our mechanistic explanation, though we do not claim formal proof.

### 4.5. Representation Shattering Generalizes to LLMs

Finally, we investigate whether our findings generalize to large Transformers trained on naturalistic data. We consider concepts with a cyclic order, in particular months of the year, and apply counterfactual edits to Llama 3 (Grattafiori et al., 2024) and Mamba S4 (Gu & Dao, 2023) (see Appx. H.1). We additionally explore non-cyclic geometries, specifically tree-structured concepts, in Appx. H.3.

In previous evaluations of KE algorithms on real-world LLMs, degradations in model performance were most consistently detectable for batched, sequential edits (Gu et al., 2024; Yoon et al., 2024). Therefore, we carry out our LLM experiments using MEMIT (Meng et al., 2023), the successor to ROME which adds support for batch editing (see Appx. E.2 for a short primer). Using MEMIT, we insert counterfactual month-order associations at varying edit distances and note how model performance and representations are affected. As for the prompts used for editing, we take inspiration from existing work by Engels et al. (2024) and use the following template: "{Month} is followed {offset} months later by the month of {}". Here, Month is uniformly sampled from {January, ⋯ , December}, and offset is uniformly sampled from {eight, nine, ten}. For edits with a counterfactual edit distance of $d$, we modify the answer to be $d$ months earlier than the true correct answer (for $d \in \{1, 2, 3\}$). For example, the ground truth answer

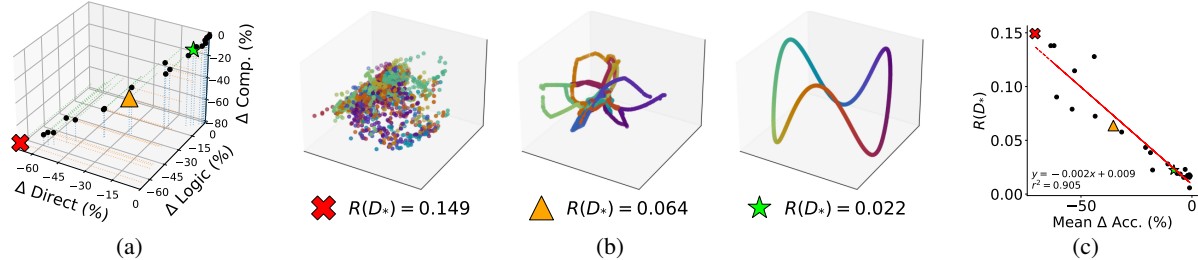

$R(D_*) = 0.149$    $R(D_*) = 0.064$    $R(D_*) = 0.022$

(a)      (b)      (c)

*Figure 5.* **Representation shattering strongly correlates with a degradation in accuracy.** *(a)* We plot the change in direct recall, logical inference, and compositional inference accuracies for different edits models, edited on different facts. We find all 3 accuracies to be strongly correlated. We select 3 edited models that span the range of accuracies, which is denoted by ✖, ▲ and ★ in the plot. *(b)* We plot the representations using a variant of Isomap (see Appx. F) with the entities colored by the cyclic order. We observe a clear trend where larger drop in accuracy directly correlates with a greater degree of representation shattering, i.e., the geometric structure of the data is destroyed after the edit. *(c)* We plot the mean drop in accuracy against the representation shattering metric $R(D_*)$ as defined in Eq. 1. Greater representation shattering is strongly correlated with more severe accuracy degradation ($r^2 = 0.905$).

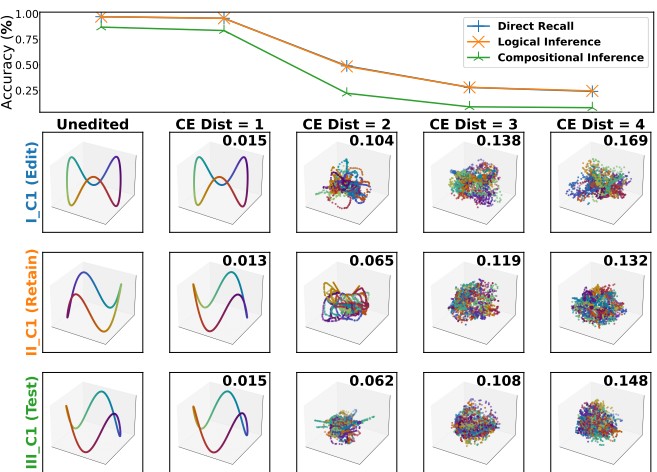

*Figure 6.* **Counterfactual edits with larger edit distance result in larger drop in accuracy and greater degree of representation shattering.** We apply counterfactual knowledge edits to overwrite a correctly learned fact (1154.I_C1=567) with inconsistent counterfactual associations. We then plot the accuracy after the counterfactual edit for different edit distances and the corresponding low-dimensional embedding of the representation obtained using a modified version of Isomap (see Appx. F). The numerical quantity in the upper right of each manifold visualization is the $R(D_*)$ value measuring the degree of representation shattering with respect to the manifold of the unedited model. Both visually and numerically, we find that a counterfactual edit with larger edit distance requires a significant distortion to the representation geometry to learn the new fact.

to the prompt "`January is followed eight months later by the month of {}`" should be "`September`"; counterfactual edits may look like changing the output to "`August`" ($d = 1$), "`July`" ($d = 2$), or "`June`" ($d = 3$). For each specific counterfactual edit distance $d$, we have (twelve months) × (three offsets) = 36 total edits—we apply these edits sequentially in batches of size 4 using MEMIT, taking care to use consistent sample orders and batch orders across all runs for fair comparisons.

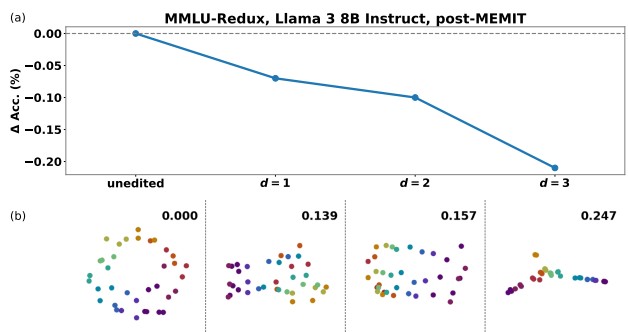

*Figure 7.* **Representation shattering also occurs in a real LLM (Llama 3 8B Instruct).** We apply KE (MEMIT) with counterfactual prompts to modify the cyclic order of calendar months in the Llama 3 8B Instruct model. After editing, we evaluate the model's ability to perform reasoning on the MMLU-Redux benchmark and visualize the model's representations for each month. *(a)* As the counterfactual edit distance grows, the model's reasoning abilities gradually decrease. *(b)* Coinciding with the degradation in reasoning performance, the gradual shattering of the ring structure in the model's intermediate representations occurs as the counterfactual edit distance is increased (activations were collected at the layer 19 output; see Appx. H.2 for other extraction points). For similar results with Mamba S4 (Gu & Dao, 2023), refer to Appx. H.1. For experiments with concepts organized in a non-circular geometry, refer to Appx. H.3.

To quantify model performance before and after editing, we adopt the MMLU-Redux reasoning benchmark (Gema et al., 2024) with the ZeroEval prompting framework (Lin, 2024) to elicit chain-of-thought reasoning.

In Fig. 7a, we examine how the reasoning performance of Llama 3 8B Instruct is affected by KE in relation to the counterfactual edit distance. We find that *as the edit distance increases, there is a gradual decline in the model's reasoning accuracy.* Furthermore, Fig. 7b shows the latent representations for the 12 months, extracted from an intermediate layer post-editing. *As the edit distance is varied from 1 to 3, the degree of representation shattering also*

*increases.* This result demonstrates that our insights from synthetic data, i.e. the representation shattering hypothesis, can generalize to larger models trained on naturalistic data.

## 5. Conclusion

In this work, we introduced a synthetic framework to analyze the side effects of knowledge editing in transformers, identifying "representation shattering" as a key factor behind performance degradation. Specifically, we show preserving representational structures underlying a model's knowledge is crucial to avoiding negative consequences of knowledge editing: distortion of such structures impacts a model's broader capabilities. To arrive at this hypothesis, we design a controlled framework that allows investigations into models modified by knowledge editing protocols, offering clear representation-level explanations for why knowledge editing can harms models' broader capabilities that generalize to real-world models like Llama 3. While the use of simplified tasks and models can limit the scope of our conclusions, since larger, more complex real-world models may exhibit additional dynamics that our framework does not capture, we believe that testing knowledge editing protocols on setups similar to our synthetic knowledge graph will significantly aid the process of designing better editing protocols. Failing even such simple, albeit systematically defined settings, likely implies the editing protocol should not be readily trusted or applied at scale.

Additionally, our study aligns with a broader theoretical framework: *(1)* Transformers store factual associations in key-value pairs within MLP layers (Geva et al., 2020; Meng et al., 2022a), corroborated by our synthetic model analysis. *(2)* Parameter sharing and superposition cluster unrelated facts in overlapping subspaces (Geva et al., 2020; Henighan et al., 2023), making them vulnerable to unintentional interference. *(3)* Entities and relations often form structured manifolds (e.g., cycles, hierarchies), which aid compositional inference (Engels et al., 2024). *(4)* KE methods (ROME, MEMIT, etc.) enact local weight updates that deform these manifolds, causing representation shattering for unedited facts residing in shared sub-regions. Points *(1)-(4)* together lead us to believe that the fragility of KE arises from the entangled, compressed nature of factual storage, rather than just from the task of knowledge retention in and of itself.

More broadly, our study suggests that mitigating the pitfalls of KE may require fundamentally rethinking the popular "localize-then-edit" paradigm. Indeed, the field has already begun moving in this direction: promising methods include Retrieval Augmentation (Lewis et al., 2020; Borgeaud et al., 2022), Lifelong Model Editing (Wang et al., 2024), and Synthetic Document Finetuning (Wang et al., 2025). We hope that our work can help guide the design of further improvements to KE approaches.

## Impact Statement

This paper introduces a scientific framework to analyze the phenomenon of representation shattering in Transformer models during Knowledge Editing. We believe the insights uncovered in this work will form a foundational step toward developing more robust and interpretable knowledge editing techniques, ultimately advancing the reliability of AI systems for societal benefit.

## Contributions

KN and ESL conceived the project direction and designed the primary experimental setup, with inputs from RR and MK. The representation shattering hypothesis was proposed by KN, ESL, and HT. KN led experiments on the synthetic knowledge graph domain. RR, KN, and ESL wrote the paper. Conceptual figures were designed by HT, with inputs from KN and MO. KN and MO conceived the natural data setup for confirming the hypothesis and ran experiments therein. KN wrote the appendix. ESL and HT supervised the project.

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

# A. Setup Details

## A.1. Source Code

Please find the source code for our experiments at github.com/KentoNishi/KE-ICML-2025.

## A.2. Pseudo-Code

Let $U(.)$ define the uniform distribution over the input. Let $X$ be the set of entities, $R$ the set of relations and $F$ the set of facts, defining a knowledge graph $G = (X, R, F)$.

```
function generateSequence()
    x_p ~ U(X) from a uniform distribution over the entities.
    S = [x_p]
    entity_flag ← False
    // Create a sequence of alternating entities and relations
    while len(S) < context_size do
        if (entity_flag) then
            // Add an entity that completes a valid fact
            Set x_n such that (x_p, r⃗, x_n) is a fact in the knowledge graph G.
            S.append(x_n)
            x_p ← x_n
        end
        else
            // Add a composition of relations
            K ~ U({1, 2}). r⃗ = [ ]
            for (i in 1 to K) do
                r ~ U(R)
                S.append(r)
                r⃗.append(r)
            end
            Set x_n such that (x_s, r⃗, x_n) is a fact in the knowledge graph G.
        end
        entity_flag ← ¬(entity_flag)
    end
    S = S[:context_size]
    return S
```

**Algorithm 1:** Generate a single sequence containing a collection of facts.

# B. Data Generation Process Details

For this study, we use the following hyperparameters for our data generation process.

- **Number of entities**: 2048
- **Number of example sequences**: $10^8$
- **Maximum composition length**: 2
- **Maximum entities per sequence**: 8

Additionally, only when generating the training dataset, the DGP may drop sequences which contain one direction of a pair of conjugate facts with fixed probability $p$. That is, for any entity $x_i$ and relations $r, r'$, suppose the fact $(x_i, r, x_j)$ always implies $(x_j, r', x_i)$ (i.e. $r = \text{I\_C1}$ and $r' = \text{I\_A1}$). The DGP may hold out *one of* these *facts* from the training data (with probability $p$). Even if $(x_j, r', x_i)$ is absent, one can still learn the conjugacy of $r$ and $r'$ and the existence of $x_i$ and $x_j$ via other examples; however, failure to infer this relation indicates the model has rote memorized relations rather than understanding the global structure. To be abundantly clear, "held-out" refers to specific facts, not entire relation tokens. In practice, we set the probability $p = \frac{2}{3}$.

## C. Prompt Formats

- **Compositional Inference:** The model is given a chain of relations (e.g., $x_i \, r_1 \, r_2$) and must produce $x_k$, with $(x_i, r_1, x_j)$ and $(x_j, r_2, x_k)$ seen separately in training but not composed. Input: ctx $x_i \, \vec{r}$ where $\vec{r} = r_1 r_2$.
- **Logical Inference:** The model is asked about a fact $(x_j, r', x_i)$ where $(x_i, r, x_j)$ was seen but $r'$ was withheld. If the model understands relation symmetry, it can infer the inverse.

## D. Model Architecture

Our Transformer model is a fork of the open-source nanoGPT repository (https://github.com/karpathy/nanoGPT). The design is inspired by GPT, and the architecture is a decode-only Transformer with a causal self-attention mask. Our hyperparameter values are as follows.

- **Batch size**: 256
- **Context length**: 16
- **Optimizer**: Adam
- **Learning rate**: $6 \cdot 10^{-4}$
- **Training epochs**: $1.5 \cdot 10^5$
- **Decay iterations**: $1.5 \cdot 10^5$
- **Momentum**: $\beta_1 = 0.9$, $\beta_2 = 0.95$
- **Activation function**: GeLU
- **Block size**: 16
- **Embedding dimensions**: 24
- **Heads**: 12

As for tokenization, we assign every entity and relation a unique token and use standard next-token prediction with cross-entropy loss. $\text{target}_n$ is the 1-shifted version of the training sequence accounting for the padding token, and $\mathbf{x}_n$ are the logit outputs of the model at the $n$th timestep.

$$\mathcal{L}(\mathbf{x}_n, \text{target } n) = -\log\left(\frac{\exp(\beta x_{n, \text{ target } n})}{\sum_{v=0}^{\#\text{tokens}} \exp(\beta x_{n,v})}\right) = -\log\left(\underbrace{\text{softmax}(\beta \mathbf{x}_n)_{\text{target } n}}_{\text{prob(target } n)}\right)$$

## E. Knowledge Editing Methods

### E.1. Rank-One Model Editing (ROME)

E.1.1. ALGORITHM DEFINITION

Rank-One Model Editing (ROME), proposed by Meng et al. (2022a), is a popular knowledge editing algorithm used on LLMs. Their contributions are two-fold: first, through "causal tracing," they find that early and mid-layer MLP modules of transformer models are implicated in encoding factual associations (see Appx. G.2.1). Second, interpreting feed-forward layers as linear associative memories encoding key-value pairs, ROME applies a rank-one update to the MLP weights.

Notationally, for a factual association $(x_i, r, x_j)$, the key is the entity $x_i$ while the value is $x_j$. In each feed-forward layer, the hidden state $\mathbf{h}_i^{(l-1)}$ at layer $l-1$ is transformed into a key $\mathbf{k}$ by the weight matrix $\mathbf{W}_{fc}^{(l)}$, and the corresponding value $\mathbf{v}$ is retrieved by the matrix $\mathbf{W}_{proj}^{(l)}$:

$$\mathbf{h}_i^{(l)} = \mathbf{W}_{proj}^{(l)} \sigma\left(\mathbf{W}_{fc}^{(l)} \mathbf{h}_i^{(l-1)}\right)$$

where $\sigma(\cdot)$ denotes the activation function.

To modify the factual association $(x_i, r, x_j)$ in the model, ROME computes a new key-value pair $(\mathbf{k}^*, \mathbf{v}^*)$, representing the entity $x_i$ and the new target entity $x_j^*$. ROME then applies a rank-one update to the weight matrix $\mathbf{W}_{proj}^{(l^*)}$ at a specific layer $l^*$ to encode this new fact:

$$\hat{\mathbf{W}}_{proj}^{(l^*)} = \mathbf{W}_{proj}^{(l^*)} + \lambda \left(\mathbf{C}^{-1}\mathbf{k}^*\right)^\top \text{ where } \lambda = \frac{\mathbf{v}^* - \mathbf{W}_{proj}^{(l^*)}\mathbf{k}^*}{\left(\mathbf{C}^{-1}\mathbf{k}^*\right)^\top \mathbf{k}^*}$$

Here, $\mathbf{C}$ is the uncentered covariance matrix of the key vectors $\mathbf{k}$, estimated by sampling tokens from a representative dataset.

The key vector $\mathbf{k}^*$ corresponds to the entity $x_i$ in the factual association $(x_i, r, x_j^*)$. The vector is computed by averaging the MLP output for $x_i$ over multiple randomly generated contexts:

$$\mathbf{k}^* = \frac{1}{N}\sum_{j=1}^{N}\sigma\left(\mathbf{W}_{fc}^{(l^*)}\gamma\left(\mathbf{a}_i^{(l^*)} + \mathbf{h}_i^{(l-1)}\right)\right)$$

where $\gamma(\cdot)$ is a normalization function, and $\mathbf{a}_i^{(l^*)}$ is the attention output at layer $l^*$.

The value vector $\mathbf{v}^*$ is optimized to maximize the model's probability of predicting the target entity $x_j^*$ given the subject $x_i$ and relation $r$. This is done by minimizing the following objective:

$$L(\mathbf{z}) = \frac{1}{N}\sum_{j=1}^{N}\left(-\log P\left(x_j^*|x_i, r\right) + D_{KL}\left(P_G\left(x_i|p'\right)||P_G\left(x_i|p'\right)\right)\right)$$

The first term maximizes the probability of the target entity $x_j^*$, while the second term controls for "essence drift" to retain information about $x_i$. This is done by sampling inputs $p'$ for which the model's outputs should not change during the edit.

### E.1.2. IMPLEMENTATION

In our implementation of ROME tailored to our model, we apply the edit at layer 1 as it is the only available early-site layer in our model configuration. The covariance matrix $\mathbf{C}$ is estimated by randomly sampling $10^5$ inputs from the validation dataset. This provides a representative set of key vectors for computing the rank-one update. To solve for the key vector $\mathbf{k}^*$, we sample $10^5$ random context sequences, with sequence lengths varying between 2 and 10 tokens. The value solver follows a similar procedure by sampling $10^2$ context sequences selected in the same manner as the key solver. The value optimization is performed using the Adam optimizer, with hyperparameters lr $= 10^{-3}$ and weight decay $= 10^{-4}$. The value solver optimizes between 5 and 500 iterations, stopping when the predicted token is replaced by $x_j^*$. The KL divergence weight is set to 3 during optimization.

### E.2. Mass-Editing Memory in a Transformer (MEMIT)

### E.2.1. ALGORITHM DEFINITION

The Mass-Editing Memory in a Transformer (MEMIT) algorithm proposed by Meng et al. (2023) generalizes ROME (Meng et al., 2022a) to inject many factual associations at once. Rather than modifying a single layer for a single key-value pair, MEMIT identifies multiple MLP layers (via causal tracing) which encode facts. For each fact $(x_i, r, x_j^*)$, a small residual $\boldsymbol{\delta}_i$ is computed to increase $P(x_j^* \mid x_i, r)$. These $\boldsymbol{\delta}_i$ vectors are then batched and distributed across the identified layers. In each layer $\ell$, let $\mathbf{K}$ and $\mathbf{M}$ respectively stack the new keys and target values derived from $\{\boldsymbol{\delta}_i\}$. Extending ROME, the weight update for layer $\ell$ is

$$\Delta^{(\ell)} = \left(\mathbf{M} - \mathbf{W}_{proj}^{(\ell)}\mathbf{K}\right)\mathbf{K}^\top\left(\mathbf{C} + \mathbf{K}\mathbf{K}^\top\right)^{-1},$$

where $\mathbf{C}$ is the uncentered covariance matrix of preexisting keys in layer $\ell$.

### E.2.2. IMPLEMENTATION

For our experiments with Llama 3 8B Instruct and Mamba 2.8B, we simply apply public implementations with off-the-shelf hyperparameters (Gupta et al., 2024a;b; Sharma et al., 2024).

# F. Visualization Methods

In Fig. 4, we demonstrated the emergence of cyclic representations within the model by extracting representations and generating 3D Isomap projections. While the visualizations support the notion that cyclical representations are present in the model, changes in the projections can be difficult to intuitively interpret due to the overlap of differently colored segments of the manifold. For example, below is a recreation of Fig. 6 using raw Isomap projections.

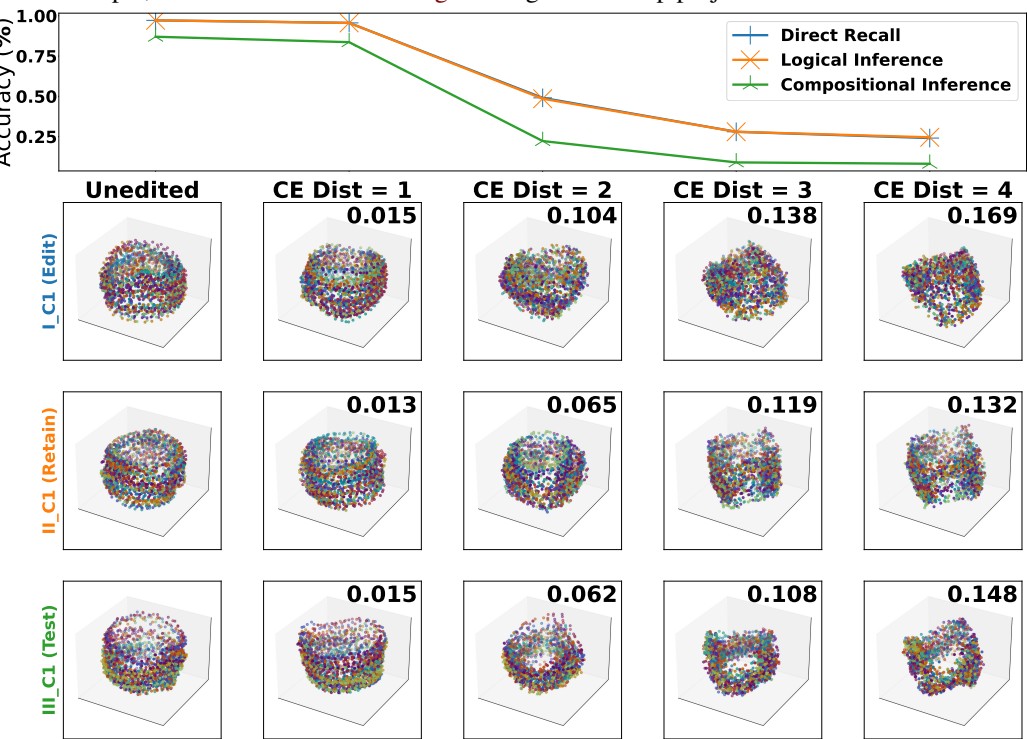

*Figure 8.* An equivalent version of Fig. 6 using the unprocessed Isomap projection renderings. Representation shattering is still visible in the flattening and clustering of points in the manifold as the counterfactual edit distance increases.

The coinciding ring segments are an artifact of the lossy projection of high-dimensional cyclical representations into a low-dimensional space: when dimensionality reduction to 3D is applied, the high-dimensional cyclical structure gets "squished" into a torus. To enhance the visual perceptibility of the representation shattering phenomenon, we additionally implement a pre-processing step to constrain the construction of the Isomap neighbors graph using the model's output predictions. More concretely, when visualizing the post-edit manifold for a particular edit $(x_i, r, x_j^*)$, we adopt the following procedure:

1. Construct a set $S_0$ of entities by prompting the *unedited* model for all immediate neighbors of $x_i$ in the cycle order of $r$ (i.e. by getting outputs for $x_i r'$ for all $r'$ in the same cycle order as $r$).
2. Apply the knowledge edit.
3. Construct a set $S_1$ of entities by collecting outputs from the *edited* model for all $s_i r$ where $s_i \in S_0$.
4. Constrain the Isomap pair-wise distance matrix to members of $S_1$.

This procedure remains faithful in comparing the pre-edit model to the post-edit model, as it relies solely on model predictions and does not introduce any ground-truth priors.

# G. Additional Synthetic DGP Experiments

## G.1. Alternative Editing Methods and Models

### G.1.1. MODEL ACCURACY

In Tab. 1, we evaluate the effects of corrective and counterfactual edits with ROME with respect to changes in the model's direct recall accuracy, logical inference accuracy, and compositional inference accuracy. The results give several key insights: corrective knowledge edits negatively affect the model's accuracy both on related and unrelated facts, intentionally introducing inconsistencies into the model's knowledge via counterfactual KE can significantly degrade model capabilities, and greater induced inconsistency (scaling the counterfactual edit distance $d$ from 1-4) causes greater performance degradation. Now, we reinforce these findings by repeating the same edits and evaluations with additional KE methods: namely **MEMIT** (Meng et al., 2023), **AlphaEdit** (Fang et al., 2024), and **PMET** (Li et al., 2024). We present our results in Tab. 3.

| KE Method | Test type | Corrective edits | | $\langle\Delta\text{Acc.}\rangle$ for Counterfactual edits | | | |
|---|---|---|---|---|---|---|---|
| | | Sub-Graph | $\langle\Delta\text{Acc.}\rangle$ | $d = 1$ | $d = 2$ | $d = 3$ | $d = 4$ |
| ROME | Direct recall | Edit | -21.95 | -01.49 | -67.01 | -77.07 | -77.94 |
| | | Retain | -22.64 | -01.91 | -66.70 | -75.49 | -75.42 |
| | | Test | -21.83 | -01.75 | -67.00 | -76.12 | -77.90 |
| | Logical inference | Edit | -22.24 | -01.44 | -67.22 | -77.14 | -78.02 |
| | | Retain | -22.50 | -01.83 | -66.88 | -75.67 | -75.67 |
| | | Test | -22.03 | -01.80 | -67.31 | -76.27 | -78.23 |
| | Compositional inference | Edit | -29.60 | -05.32 | -73.15 | -80.35 | -80.63 |
| | | Retain | -31.92 | -05.32 | -71.21 | -78.70 | -78.87 |
| | | Test | -31.70 | -06.69 | -74.88 | -81.38 | -80.62 |
| MEMIT | Direct recall | Edit | -09.51 | -01.64 | -57.98 | -67.04 | -68.72 |
| | | Retain | -07.08 | -01.78 | -48.68 | -57.23 | -58.52 |
| | | Test | -06.54 | -01.19 | -51.85 | -63.96 | -70.26 |
| | Logical inference | Edit | -09.58 | -01.61 | -58.16 | -67.31 | -69.10 |
| | | Retain | -06.73 | -01.64 | -48.45 | -57.55 | -58.66 |
| | | Test | -06.67 | -01.37 | -52.37 | -64.65 | -70.99 |
| | Compositional inference | Edit | -11.43 | -01.85 | -57.79 | -67.82 | -71.79 |
| | | Retain | -08.34 | -00.68 | -53.05 | -62.71 | -64.09 |
| | | Test | -10.47 | -03.30 | -53.36 | -66.81 | -73.42 |
| AlphaEdit | Direct recall | Edit | -06.05 | -01.45 | -54.68 | -64.01 | -63.48 |
| | | Retain | -04.68 | -01.69 | -43.72 | -52.36 | -53.63 |
| | | Test | -03.75 | -00.92 | -47.53 | -59.57 | -66.09 |
| | Logical inference | Edit | -06.13 | -01.42 | -54.93 | -64.42 | -63.91 |
| | | Retain | -04.37 | -01.55 | -43.58 | -52.74 | -53.93 |
| | | Test | -03.85 | -01.03 | -48.05 | -60.38 | -66.83 |
| | Compositional inference | Edit | -07.75 | -01.72 | -55.82 | -66.42 | -68.35 |
| | | Retain | -05.99 | -00.08 | -50.19 | -59.62 | -61.57 |
| | | Test | -07.03 | -02.75 | -51.14 | -64.14 | -70.95 |
| PMET | Direct recall | Edit | -03.97 | -01.34 | -48.27 | -50.80 | -54.72 |
| | | Retain | -02.78 | -01.61 | -35.54 | -39.18 | -46.36 |
| | | Test | -02.01 | -00.98 | -43.40 | -44.29 | -52.67 |
| | Logical inference | Edit | -04.02 | -01.32 | -48.48 | -51.05 | -55.06 |
| | | Retain | -02.47 | -01.47 | -35.40 | -39.39 | -46.60 |
| | | Test | -02.10 | -01.11 | -44.07 | -44.76 | -53.32 |
| | Compositional inference | Edit | -05.60 | -01.37 | -49.89 | -55.65 | -60.62 |
| | | Retain | -03.09 | -00.23 | -42.24 | -47.87 | -53.78 |
| | | Test | -04.56 | -02.95 | -47.00 | -50.95 | -58.98 |

*Table 3.* Results of Tab. 1, replicated using MEMIT (Meng et al., 2023), AlphaEdit (Fang et al., 2024), and PMET (Li et al., 2024). Overall, recent methods succeeding ROME are slightly less damaging to model accuracy. However, all evaluated methods nonetheless cause undesirable performance degradations in similar ways to ROME (especially for increased counterfactual edit distances). This suggests that KE methods, despite their differences in approaches, often suffer from similar shortcomings in terms of negatively impacting model performance.

### G.1.2. REPRESENTATION SHATTERING METRIC

In Tab. 2, we showed that increasing the distance of the counterfactual edit results in an increase in the extent of shattering, as numerically captured by $R(D_*)$. In similar spirit to Appx. G.1.1, we seek to verify whether this relationship between counterfactual edit distance and representation shattering holds for methods other than ROME, i.e. MEMIT (Meng et al., 2023), AlphaEdit (Fang et al., 2024), and PMET (Li et al., 2024). We present our results in Tab. 4.

| Method | Sub-Graph | $d = 1$ | $d = 2$ | $d = 3$ | $d = 4$ |
|---|---|---|---|---|---|
| ROME | Edit | 01.80 | 21.93 | 26.22 | 27.90 |
| | Retain | 01.80 | 20.84 | 25.32 | 27.28 |
| | Test | 01.84 | 21.89 | 26.52 | 28.68 |
| MEMIT | Edit | 01.89 | 08.58 | 09.32 | 08.78 |
| | Retain | 01.86 | 07.31 | 07.66 | 07.50 |
| | Test | 01.85 | 07.49 | 08.35 | 07.70 |
| AlphaEdit | Edit | 01.86 | 07.77 | 08.44 | 07.68 |
| | Retain | 01.85 | 06.51 | 06.89 | 06.99 |
| | Test | 01.83 | 06.89 | 07.60 | 06.99 |
| PMET | Edit | 01.83 | 06.55 | 06.44 | 06.41 |
| | Retain | 01.84 | 05.45 | 05.42 | 05.85 |
| | Test | 01.83 | 06.14 | 05.75 | 06.31 |

*Table 4.* Results from Tab. 2, replicated using the alternative knowledge editing methods of MEMIT (Meng et al., 2023), AlphaEdit (Fang et al., 2024), and PMET (Li et al., 2024). These successors to ROME achieve lower amounts of representation shattering overall, coinciding with their more favorable performance in Appx. G.1.1. However, the relationship between greater counterfactual edit distance $d$ and greater representation shattering $R(D_*)$ still robustly holds for all methods. This result again shows that various KE methods struggle in similar ways: specifically, the greater the inconsistency between the model's original knowledge and the edited fact, the greater the resulting distortion upon the model's representations.

## G.2. Validation of KE Algorithms' Assumptions

The knowledge editing methods we test across our experiments—namely ROME (Meng et al., 2022a), MEMIT (Meng et al., 2023), AlphaEdit (Fang et al., 2024), and PMET (Li et al., 2024)—make assumptions about the mechanisms by which language models store factual associations in their learned weights. Here, we verify those assumptions to ensure that our application of KE to our synthetic experimental setup is in fact appropriate.

### G.2.1. CAUSAL TRACING OF FACTUAL ASSOCIATIONS

ROME (Meng et al., 2022a) and its follow-up derivatives rely on "causal tracing" to locate facts within the parameters of large pretrained autoregressive transformer models. This is essentially causal mediation analysis applied to the internal states of transformer models, with scores attributing each state's contribution toward a correct factual prediction. KE methods build upon the finding that, early/middle MLP layers play a decisive role in recalling key-value facts about the subject (Meng et al., 2022a). Below, we investigate whether this finding also generalizes to our toy model by performing causal tracing.

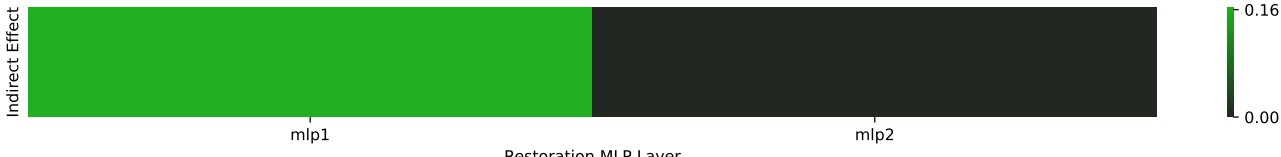

*Figure 9.* The average "Indirect Effect" of MLP output states (for the last subject token) on the final fact output elicited from the model, computed across 100 randomly sampled prompts. While our model only has two layers, causal tracing attributes higher scores to the first MLP layer rather than the second. This aligns with the findings of Meng et al. (2022a) who showed that early and mid-layer MLPs play a larger role in factual recall than later sites. We therefore target the first layer MLPs for our knowledge editing experiments.

### G.2.2. TRANSFORMER MLPS AS LINEAR ASSOCIATIVE MEMORY

Prior works view the $\mathbf{W}_{proj}^{(l)}$ matrix of the MLP layer within the Transformer architecture as linear associative memory. Under this perspective, the MLP layers act as key-value stores for vector keys and corresponding vector values (Kohonen, 2009; Anderson, 1972; Geva et al., 2020; Bau et al., 2020). Below, we qualitatively investigate whether this observation also holds for facts learned by our toy model.

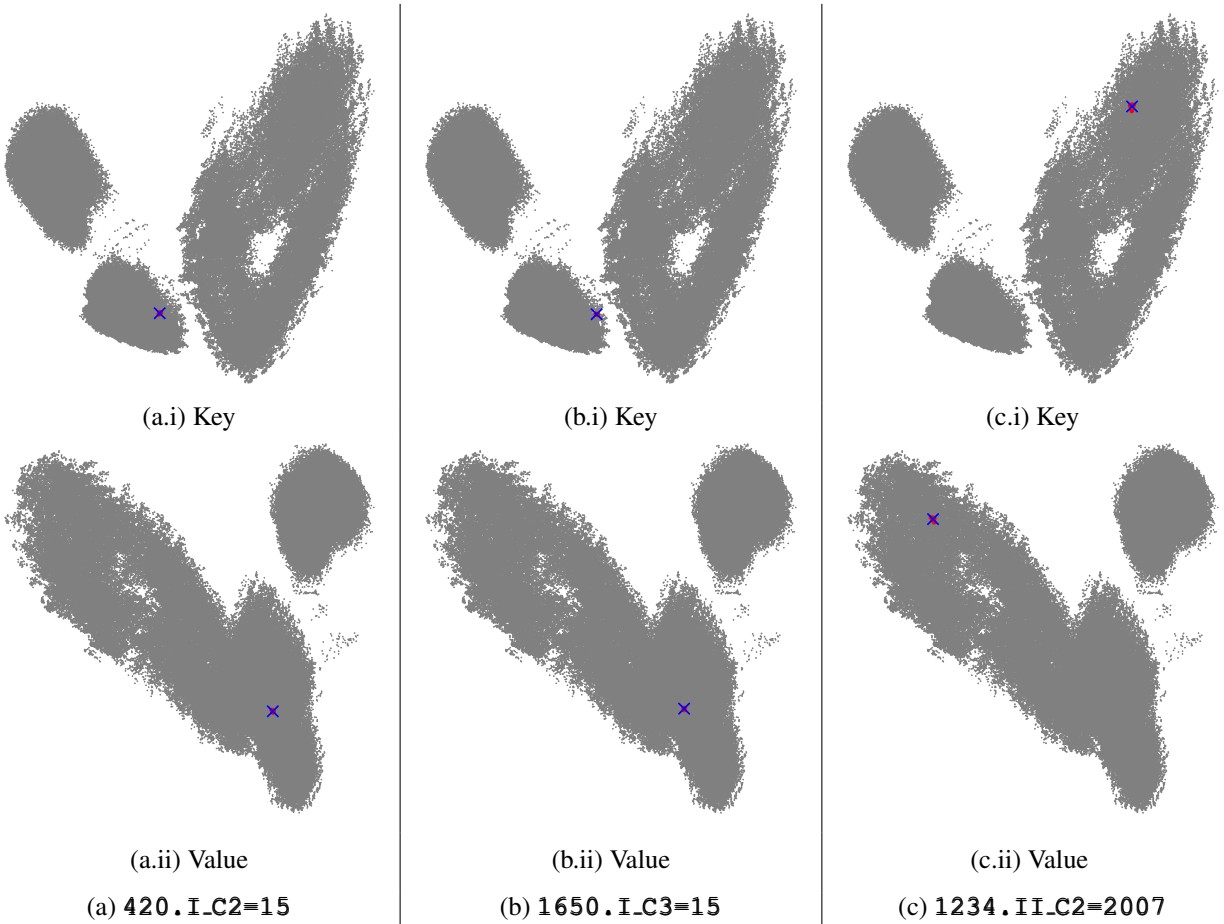

|       (a.i) Key       |       (b.i) Key       |       (c.i) Key       |
| :-------------------: | :-------------------: | :-------------------: |
|      (a.ii) Value     |      (b.ii) Value     |      (c.ii) Value     |
| (a) `420.I_C2=15`     | (b) `1650.I_C3=15`    | (c) `1234.II_C2=2007` |

*Figure 10.* Inputs (keys) and outputs (values) of the layer 1 MLP module, projected to 2D with PCA. Gray corresponds to randomly sampled prompts across all possible facts, red to randomly sampled prompts for a specific fact, and blue to the average position of the red points. Comparing *(a)* and *(b)*, we find that fact prompts which resolve to the same final target entity in the same subgraph (i.e., entity `15`) share similar keys and values. Meanwhile, a completely different fact (like `1234.II_C2=2007`) is stored as a clearly distinct key-value pair. These patterns support the key-value framework as compatible with our synthetic setup.

### G.3. Independence of Cyclic Orders

In our evaluations, we make edits to various relations under the assumption that the Transformer internalizes the independence of the cyclic orders (`I`, `II`, and `III`). Here, we ask: do the model's internal representations truly reflect this? We answer this question by inspecting the representations for the output of the multi-head attention output in layer 2 at the last token position using PCA. Unlike in previous sections where we focused on a fixed relation $r$ and varied $x_i$ for inputs of the form $\cdots x_i r$, we now vary both $x_i$ and $r$ and color-code each projection by the cyclic order to which the relation $r$ belongs. We present the resulting projections in Fig. 11, and find that prompts eliciting knowledge for each cyclic order are clustered closely together in the latent space—this is further evidence that the model internalizes the properties of the underlying knowledge graph.

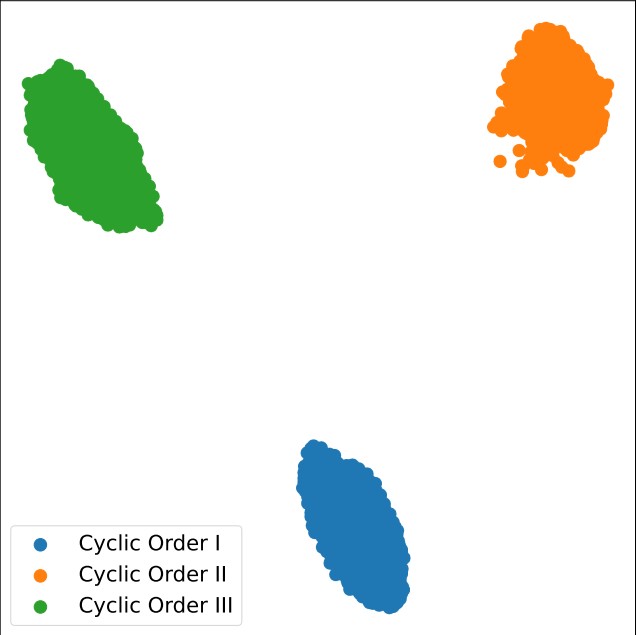

*Figure 11.* PCA of representations extracted from the output of the multi-head attention output in layer 2 at the last token position, color-coded by the cyclic order of the last relation token.

## G.4. Manifolds for Various Representation Extraction Points

We repeat our representation visualizations analysis for all relations at different layers in the model and at different sequence positions, finding the structured representations are found at specific token positions. See Fig. 12.

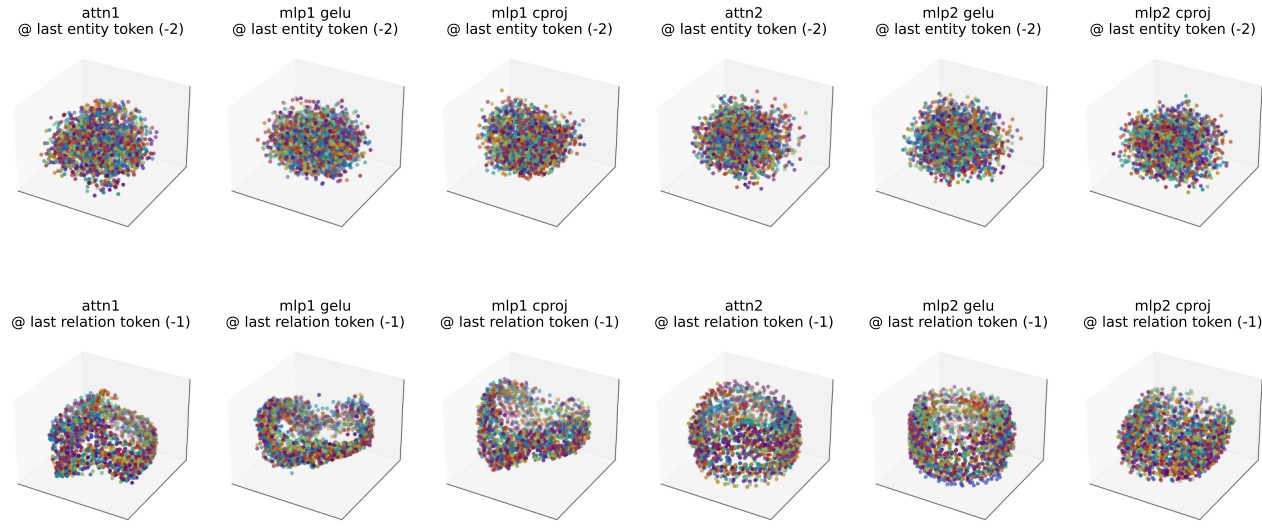

*Figure 12.* 3D Isomap projections for representations extracted from various token positions and points in the model. The cyclical representation manifolds can only be observed for the last relation token position ($-1$th token), and not at the last entity token position ($-2$th token). This intuitively makes sense because the last relation token informs the model about which cycle order the current input is querying for. We primarily use the "attn2 last relation token" representations throughout this work because it is the earliest point at which a well-structured cyclical manifold can be observed beyond the point of the ROME intervention (which is at "mlp1 cproj").

## G.5. Manifolds for All Relations

In Fig. 13, we provide isomap projections of representations extracted for all relations from our model. We show highly structured representations are formed within the model, indicating the model is truly learning the data-generating process and not merely memorizing information.

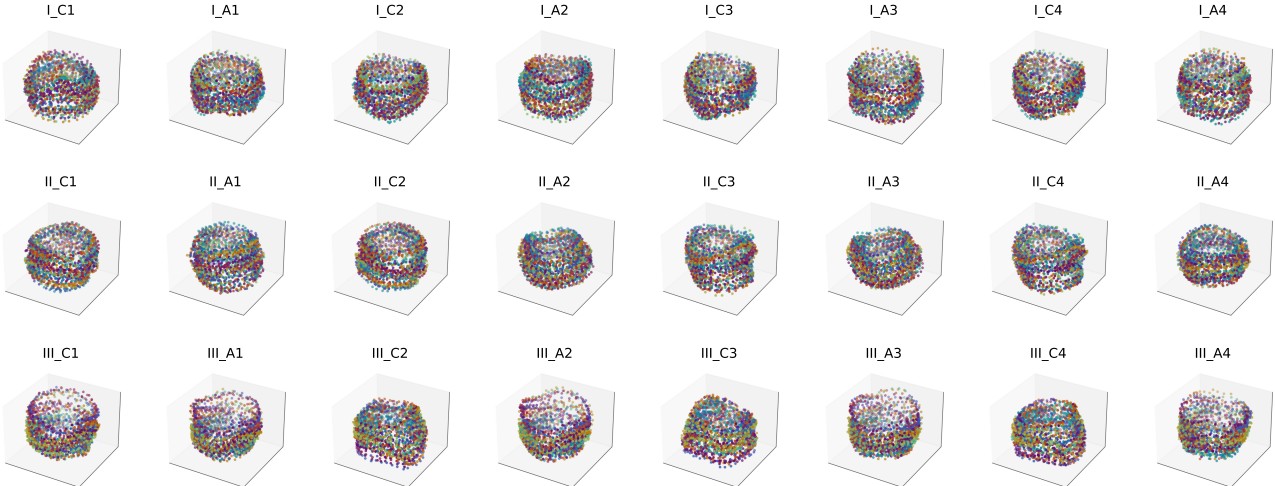

*Figure 13.* Isomap projections for representations for all relations, extracted from the output of the multi-head attention output in layer 2 at the last token position. We find that all relations are represented by a cyclical representation manifold. This shows that the model is not falling back on memorization for any relations—rather, it represents all of its knowledge in consistent, ring-like manifolds.

## G.6. Counterfactual Editing

### G.6.1. DISTRIBUTION OF DEGREDATIONS FOR COUNTERFACTUAL EDITS

The plots in Fig. 14 correspond to the counterfactual editing results presented in Sec. 4.4 and Tab. 1.

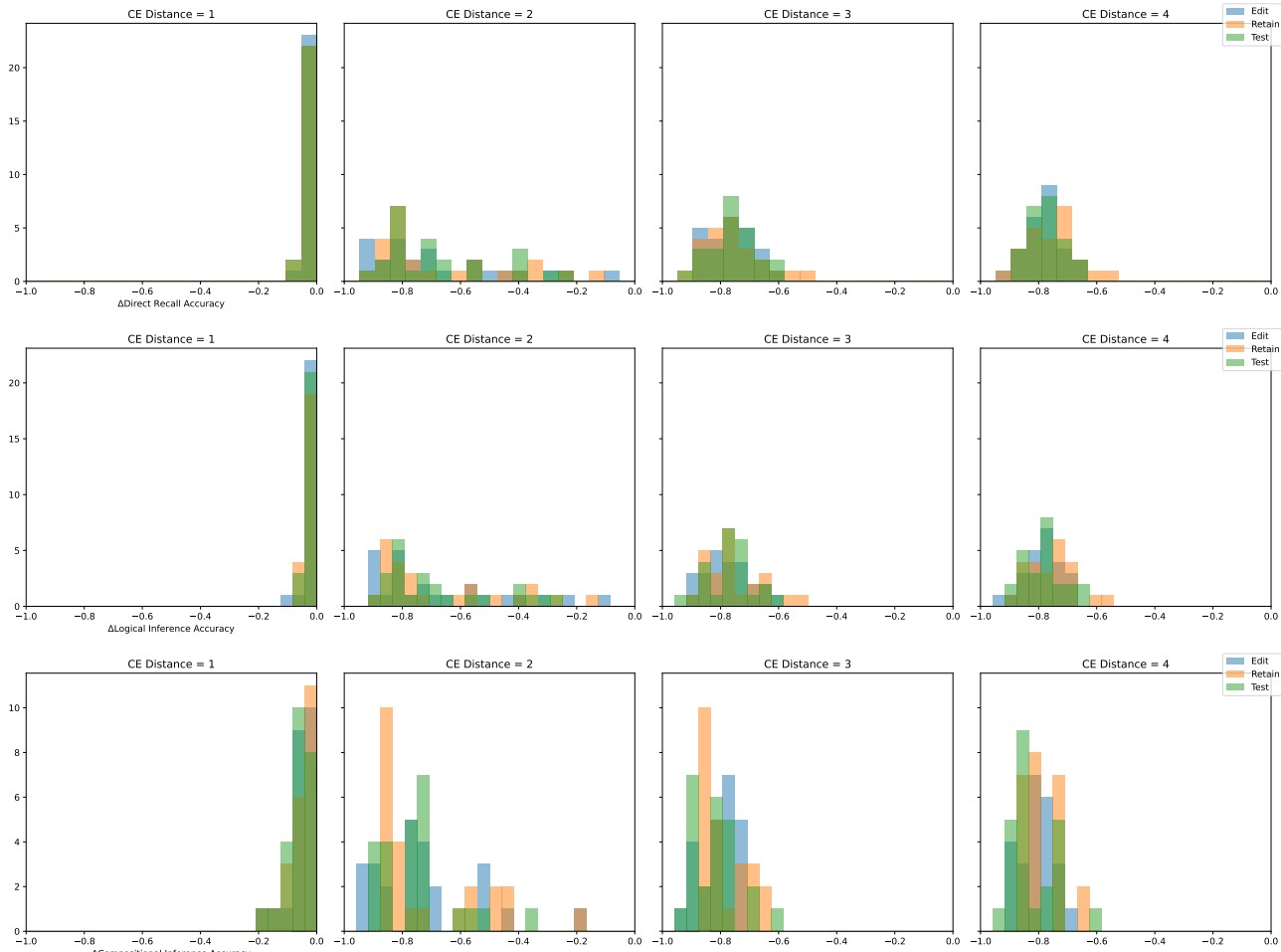

*Figure 14.* Distribution of post-edit accuracy degredations for direct recall, logical inference, and compositional inference in relation to the counterfactual edit distances. A significant shift can be observed between CE distances of 1 and 2, showing the point at which detrimental representation shattering can occur.

### G.6.2. ADDITIONAL VISUALIZATIONS

In Fig. 6, we showcase an example of the change in accuracies and representation manifolds when applying a counterfactual edits (specifically for fact `1154.I_C1`). For a more representative view, we additionally provide more examples of counterfactual edits (with both raw and pre-processed versions side-by-side, as described in Appx. F).

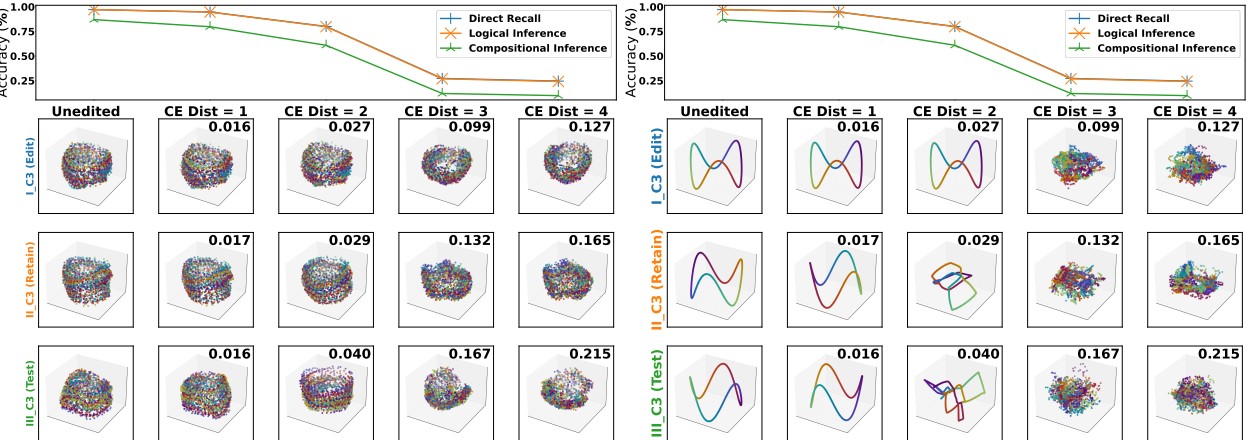

*Figure 15.* Counterfactual editing visualizations for `1623.I_A2`.

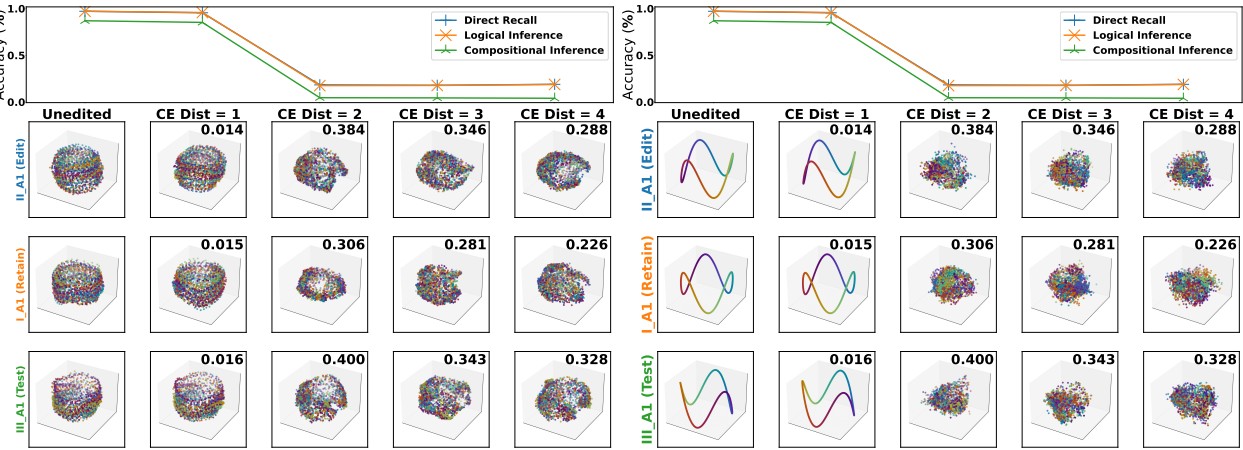

*Figure 16.* Counterfactual editing visualizations for `1121.II_C1`.

# H. Additional Naturalistic LLM Experiments

## H.1. Knowledge Editing on Mamba

In Sec. 4.5, we investigate whether the representation shattering hypothesis generalizes to large Transformers trained on naturalistic data. We consider the cyclic order of the months of the year and apply counterfactual edits to Llama 3 8B Instruct (Grattafiori et al., 2024) and found that as we vary the edit distance from 1 to 3, the degree of representation shattering increases. To further probe the robustness of our claims with respect to model size and model architecture, we additionally explore KE with Mamba (Gu & Dao, 2023). Mamba is a structured state space sequence model, and we use the Mamba-2.8B variant for this experiment. For consistency with the Llama experiments, we use the MEMIT editing method adapted appropriately to work with the Mamba architecture (as per Sharma et al. (2024)). For the counterfactual edit prompts, we use the same prompts as in Sec. 4.5 (i.e. "`{Month} is followed {offset} months later by the month of {}`").

We present the resulting manifold visualizations and $R(D_*)$ values in Fig. 17. We find that the relationship between the counterfactual edit distance and the extent of representation shattering can be reproduced in Mamba 2.8B, much like in Llama 3 8B Instruct. As a side note, we note that benchmarking Mamba 2.8B on MMLU-Redux (Gema et al., 2024) was not achievable since Mamba 2.8B is not instruction-tuned; nonetheless, the gradual shattering of representations in Mamba 2.8B induced by larger counterfactual edit distances suggests that the representation shattering hypothesis is generally applicable across different model types and architectures.

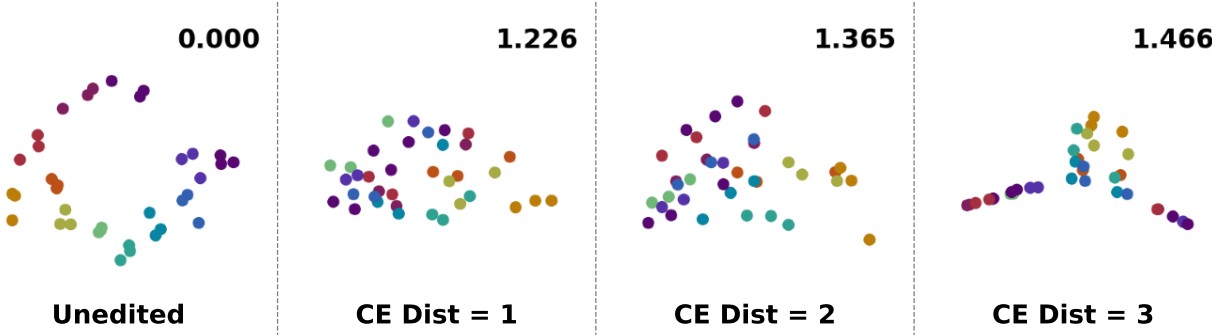

*Figure 17.* Fig. 7, replicated using Mamba 2.8B (Gu & Dao, 2023). Like in Llama 3 8B Instruct, the ring structure shatters for larger counterfactual edit distances (activations were collected at the layer 38 output; see Appx. H.2 for other extraction points). This result demonstrates that our findings are not limited to specific models and can be extended to various architectures.

## H.2. LLM Representation Extraction Points

To measure and visualize representation shattering in Sec. 4.5 and Appx. H.1, we extracted intermediate representations from Llama 3 8B Instruct and Mamba 2.8B at the output of layers 19 and 38, respectively. We chose these layers because they were the earliest points in each model for which cyclic representations could be observed at the last token position (as shown in Fig. 18 and Fig. 19).

**Llama 3 8B Instruct**

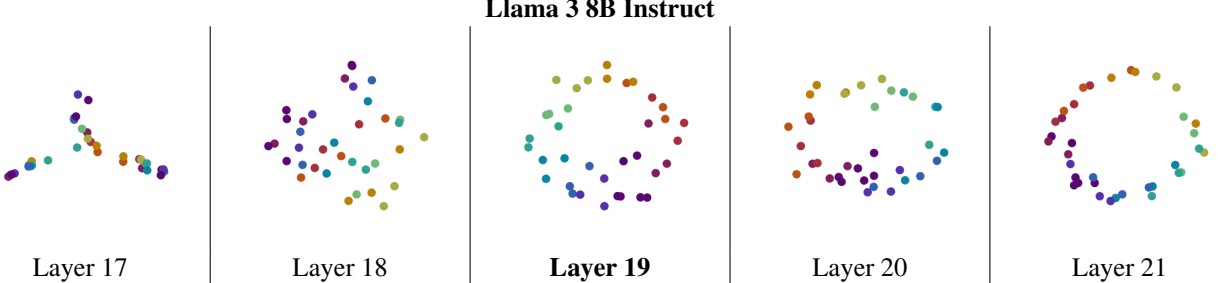

| Layer 17 | Layer 18 | **Layer 19** | Layer 20 | Layer 21 |

*Figure 18.* Representations extracted for the months of the year in Llama 3 8B Instruct, at the outputs of layers 17-21. We select layer 19 for our analysis of representation shattering because it is the earliest layer at which a clear cyclic structure is observed.

**Mamba 2.8B**

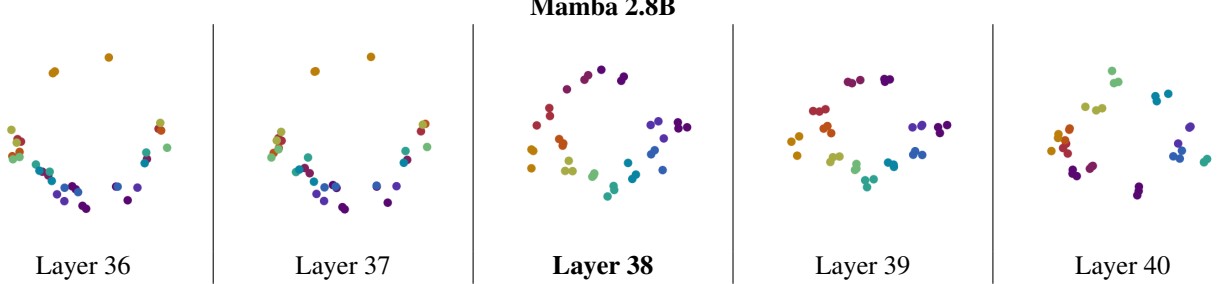

| Layer 36 | Layer 37 | **Layer 38** | Layer 39 | Layer 40 |

*Figure 19.* Representations extracted for the months of the year in Mamba 2.8B, at the outputs of layers 36-40. We select layer 38 for our analysis of representation shattering because it is the earliest layer at which a clear cyclic structure is observed.

## H.3. Knowledge Editing with Naturalistic Trees

In our experiments, we primarily focus on synthetic knowledge graphs with cyclical structures. While the simplicity of cycles is desirable for our synthetic experiments, real human knowledge and language can exhibit more complex structures. For example, geographical ground-truths can be expressed in a tree structure, with entities like cities/countries/continents having relations with other cities/countries/continents, i.e. $x_i = $ Paris, $r = $ located in country, $x_j = $ France.

Here, we ask: does the representation shattering hypothesis hold for more realistic tree-shaped knowledge graphs in more complex LLMs? To answer this question, we take inspiration from the classic "The Eiffel Tower is located in the city of Rome" example of counterfactual knowledge editing (Meng et al., 2022a). For our purposes, we edit the country associations of major cities. In particular, we consider the following five countries: *France*, *Spain*, *Italy*, *Germany*, and *the United Kingdom*. Then, we also consider the five most populous cities of each country, totaling 25 cities: *Paris*, *Marseille*, *Lyon*, *Toulouse*, *Nice*, *Madrid*, *Barcelona*, *Valencia*, *Sevilla*, *Zaragoza*, *Rome*, *Milan*, *Naples*, *Turin*, *Palermo*, *Berlin*, *Hamburg*, *Munich*, *Köln*, *Frankfurt am Main*, *London*, *Birmingham*, *Liverpool*, *Glasgow*, and *Sheffield*. The knowledge graph involving these city-country pairs contains facts such as ($x_i = $ Paris, $r = $ located in country, $x_j = $ France). The ground-truth arrangements of the cities and countries form a tree (Fig. 20a).

From the latent space of LLMs, however, it is difficult to extract clean tree-like geometries. When we project the representations for tokens corresponding to the country and city names using Isomap, the result does not yield a discernible tree shape (Fig. 20b). Despite the exact structure of the latent space not being clear, the notion of "distance" in the manifold can still be applied. For example, in Fig. 20b, *Spain* is closer to *France* than is *the United Kingdom*; therefore, the edit "Paris is a city in the country of Spain" has a smaller counterfactual edit distance than does the edit "Paris is a city in the country of the United Kingdom." Fig. 21a and Fig. 21b show the representation manifold Isomaps after applying the edits "Paris is a city in the country of Spain" and "Paris is a city in the country of the United Kingdom," respectively, using ROME on GPT-2 XL. First, we find that both counterfactual edits cause the representations for all cities and countries to collapse inward. Moreover, the edit to "the United Kingdom" causes a greater distortion than the edit to "Spain," as is evident both by visual inspection and by the numerical representation shattering quantity $R(D_*)$.

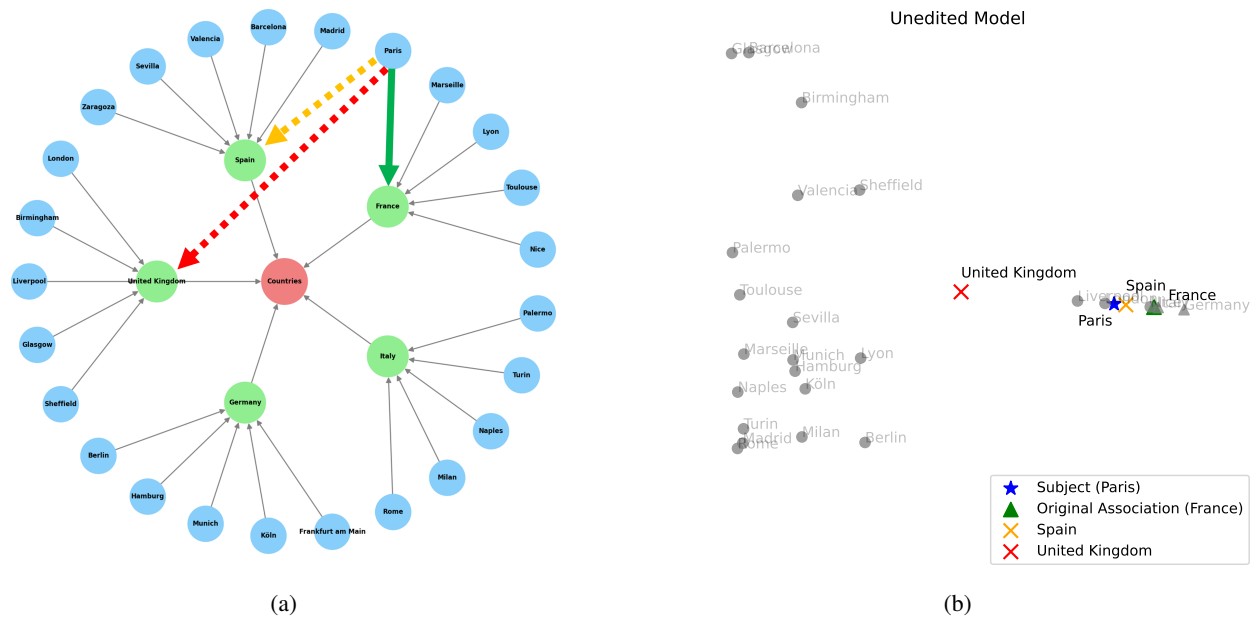

(a)                                                          (b)

*Figure 20. (a)* The ground-truth tree representing the 5 countries and its 25 cities. The correct factual association for the prompt "Paris is a city in the country of..." is France. In this example, we consider the counterfactual edits "Paris is a city in the country of Spain" and "Paris is a city in the country of the United Kingdom". *(b)* Isomap projections of representations for the selected countries and cities. We find that, on this model's representation manifold, editing Paris to be in Spain constitutes a smaller counterfactual edit distance than does editing Paris to be in the United Kingdom.

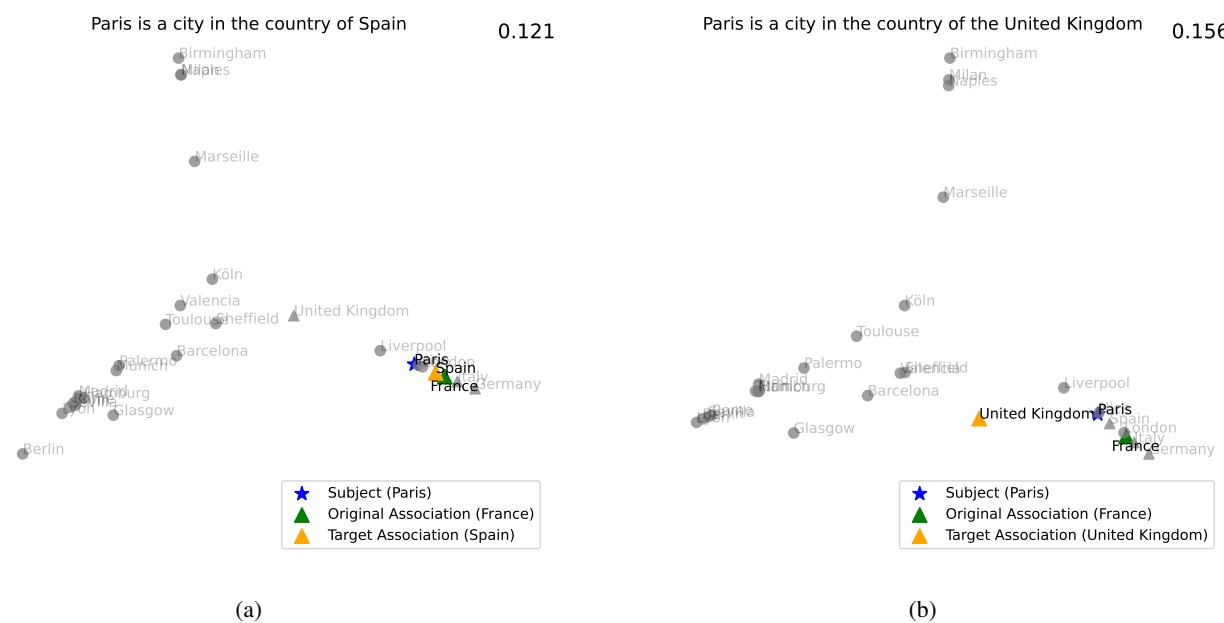

*Figure 21.* Isomap projections of latent representations after applying a counterfactual edit. *(a)* "Paris is a city in the country of Spain." *(b)* "Paris is a city in the country of the United Kingdom."

To take a step in verifying whether this finding is generalizable, we applied counterfactual edits to each of the 25 selected cities. For each city, we computed the country which constitutes the "closest" and "furthest" counterfactual edit distance on the model's representation manifold. After applying the two counterfactual edits, we computed $R(D_*^{\text{farthest}})$ and $R(D_*^{\text{closest}})$. Across the 25 cities, the average ratio $R(D_*^{\text{farthest}})/R(D_*^{\text{closest}})$ was $1.1483$. In other words, when changing a city's parent country, editing to a close country on the representation manifold yields less shattering than editing to a country which sits far away on the manifold.

These preliminary results align with our main hypothesis: KE methods distort language models' representations in order to insert new facts or alter old ones (i.e. representation shattering), and the extent of representation shattering increases with the distance between the old fact and the desired new fact on the manifold.

