# OpenReview forum: "Representation Shattering in Transformers: A Synthetic Study with Knowledge Editing"
_ICML.cc/2025/Conference — ICML 2025 poster_

### Official Review · Reviewer_kjUD · 2025-03-08

**Overall Recommendation:** 4

**Summary:**

This paper proposes a synthetic framework to investigate knowledge editing in Transformer-based language models. The authors learn a model from structured knowledge out of a graph featuring multiple cyclic orders of entities and relations of facts. By evaluating representation changes following targeted edits to specific facts, the authors identify a phenomenon termed “representation shattering”: editing even a single fact can significantly distort the learned geometric manifold of entity embeddings, and decrease factual recall and reasoning performance. They further demonstrate that this decrease increases with the geometric distance or counterfactuality of edits, where edits that more drastically conflict with the previously learned geometry cause greater disruption. Finally, the synthetic findings are validated through experiments on real LLMs (Llama and Mamba) by editing the cyclic ordering of months, where edits inducing greater distortion lead to worse performance on benchmarks.

**Claims And Evidence:**

The authors claim that models acquire coherent representations of relational data, effectively capturing the underlying structural geometries. To this end, they use cyclic graphs. Using dimensionality reduction (Isomap) visualizations, they illustrate that entity embeddings naturally form clear ring-shaped geometries aligned with the cyclic structure of the data. The models also exhibit strong performance in logical and compositional inference tasks, indicating that they internalize structural patterns beyond simple memorization of isolated facts.

Additionally, the authors argue that knowledge editing methods (such as ROME and MEMIT) inadvertently harm performance on unrelated facts and reasoning tasks, extending well beyond the specific fact being edited. They systematically quantify these accuracy reductions across direct recall (facts seen during training), logical inference (such as relation reversals), and compositional inference (multi-hop reasoning), consistently finding that edits negatively impact accuracy even for unrelated information.

The underlying cause of these degradations is identified as “representation shattering”: forced edits distort the learned embedding geometry. Through visualizations and quantitative measures, the authors demonstrate that structural disruption increases with the magnitude of edits. They introduce a metric, R(D^*), to measure the degree of embedding shifts post-editing, revealing a strong correlation between embedding distortions and performance deterioration.

Furthermore, the authors demonstrate that these findings generalize to real-world large language models (e.g., Llama 3 and Mamba S4). Specifically, edits applied to the cyclic ordering of months in real models yield similar representational distortions and degraded performance on external benchmarks (MMLU-Redux), echoing the synthetic experimental results.

Overall, I find these claims are supported by both synthetic experiments and validations on real models, evaluating changes in representation as observed performance losses.

**Essential References Not Discussed:**

I find the relevant related work sufficiently discussed.

**Experimental Designs Or Analyses:**

The synthetic dataset is systematically constructed to cover direct, logical, and compositional facts. The authors vary the distance of counterfactual edits in the cyclic order to systematically test how conflict magnitude affects distortions to analyze edit distance. They replicate their findings on Llama 3 and Mamba S4, focusing on month ordering as a real-world example. The main limitation is that these evaluations are primarily small-scale or specifically for months, though the authors mention an extension to tree-structured data in the appendix. Still, I find the analysis consistent and well-motivated.

Overall, the experimental methodology is appropriate and the analyses appear valid. The results are well-documented, showing strong negative correlations between manifold distortion and model performance.

**Methods And Evaluation Criteria:**

Method:
The paper sets up a cyclic synthetic knowledge graph and creates various k-hop relations (e.g., 2-hop clockwise neighbor) to form several subgraphs (edit, retain, test). This framework appears carefully designed to mimic real-world relational data while being simple enough to visualize and measure precisely.

Evaluation:
The authors compute (1) direct recall, (2) logical inference, and (3) compositional inference accuracy. This is appropriately motivated: they state that knowledge editing must preserve (or at least minimally disrupt) both memorized facts and the underlying relational inferences.
The authors define a Frobenius norm–based measure to quantify how much embeddings shift before versus after an edit. This is a simple but intuitive way to measure global manifold changes.

I find both the evaluation and methodology are reasonable for the approach.

**Other Comments Or Suggestions:**

Exploring connections to classical catastrophic forgetting mitigation techniques, such as mentioned in the related work (e.g., elastic weight consolidation, low-rank updates) as the author is mentioned for future work, might yield useful insights. Also a more fine-grained analysis of multi-step and layer-wise edits would be interesting.

**Other Strengths And Weaknesses:**

Strengths:
While the idea is not entirely novel and bills and work from interpretability research, the author's present a clearly structured framework with multiple inference types and evaluate embedding distortions with performance degradation. The paper is well-structured and as such clear to read. The step from synthetic to realistic scenarios in real-world models underpins the approach.

Weaknesses:
While a realistic scenario is evaluated, the exploration in month ordering remains limited. The focus lies on cyclic structures, with minimal emphasis on broader relational types, which would be interesting to gain more insights and also gain more insights to mitigate representational shattering.

**Questions For Authors:**

How would you expect the approach to generalize beyond cycles? Similarly, how would you expect the approach to work on more fine-grained tasks than the month ordering?

**Relation To Broader Scientific Literature:**

The paper relates to existing knowledge editing approaches, such as ROME and MEMIT. It highlights common pitfalls like catastrophic forgetting and representational contradictions, such as the ones from [1], [2], [3].



The work relates directly to existing studies on how Transformers encode and manipulate structured conceptual representations, such as [4] who explore interpretations of feed-forward layers as key-value memory stores and [5], who aim to contextualize why local edits (e.g., rewriting a single fact) influence a model’s stored knowledge.

The synthetic dataset generation approach relates to previous work, such as [6] investigating compositional reasoning and interpretability in Transformer models.

Overall, I find the authors do a good job covering related work to ground their analysis.

[1] Hase, P., Bansal, M., Kim, B., and Ghandeharioun, A. Does Localization Inform Editing? Surprising Differences in Causality-Based Localization vs. Knowledge Editing in Language Models. arXiv preprint arXiv:2301.04213
[2] Gu, J.-C., Xu, H.-X., Ma, J.-Y., Lu, P., Ling, Z.-H., Chang, K.-W., and Peng, N. Model Editing Harms General Abilities of Large Language Models: Regularization to the Rescue. arXiv preprint arXiv:2401.04700, 2024.
[3]Cohen, R., Biran, E., Yoran, O., Globerson, A., and Geva, M. Evaluating the Ripple Effects of Knowledge Editing in Language Models. arXiv preprint arXiv:2307.12976
[4] Geva, M., Goldberg, Y., & Berant, J. (2020). Transformer feed-forward layers are key-value memories. In Proceedings of the 2020 Conference on Empirical Methods in Natural Language Processing (EMNLP).
[5] Allen-Zhu, Z. & Li, Y. (2023). Physics of language models
[6] Compositional Abilities Emerge Multiplicatively: Exploring Diffusion Models on a Synthetic Task.

**Theoretical Claims:**

The paper does not provide formal proofs but a mechanistic viewpoint relying on the hypothesis that transformer models embed related entities in geometric manifolds and that forcibly altering facts shifts these embeddings leading to general performance drops. This motivations and also demonstrations in controlled settings lend credibility to their hypothesis.

Given the nature of the paper, there are no deep theorems or long proofs to check. The “representation shattering” phenomenon is presented more as an empirical discovery and conceptual explanation than a formal claim that requires a proof. Hence, there are no apparent issues with unverified theoretical claims.

⸻

---

> ### Author Rebuttal · Authors · 2025-03-31
>
> We thank the reviewer for appreciating our empirical analysis and mechanism of representation shattering. Please see our detailed responses to specific comments below.
>
> ---
>
> > **#1: Multi-Step Editing**
>
> Thank you for this suggestion! We note we already provide an analysis of multi-step and layerwise edits with real LLMs (Llama 3 and Mamba in Sec. 4.5 and App. G.1) via MEMIT:
>
> > For each counterfactual edit distance $d$, we have $(\text{twelve months}) \times (\text{three offsets}) = 36$ total edits—we apply these sequentially in batches of size $4$ using MEMIT, using consistent sample orders and batch orders for fair comparisons.
>
> This was done to magnify performance degradations and representation shattering, as singular edits yielded minute effects initially. This qualitatively suggests shattering amplifies in multi-step editing, but a rigorous comparison of Batch vs. Sequential vs. Sequential-Batched editing (as in [1]) is warranted. We will incorporate such comparisons into future work.
>
> [1] Yoon et al., 2024
>
> ---
>
> > **#2: Layerwise Analysis**
>
> Current KE methods (ROME/MEMIT) specify early layers (typically early-layer MLPs) as edit sites based on causal tracing. Altering this would require formulating and validating a new KE approach, beyond our scope. We do note that we include analyses in App. F.3 and G.2 visualizing representations from various model sites, hence showing the effects of KE across layers.
>
>
> ---
>
> > **#3: Connections to Catastrophic Forgetting Mitigation**
>
> Thank you for highlighting connections to forgetting mitigation strategies! Indeed, our framework can naturally evaluate such strategies: effective forgetting mitigation methods should preserve global representation geometry when instilling new knowledge (minimizing $R(D_*)$). Recent work in fact examines regularization-based KE methods (e.g., [1]), suggesting compatibility with classical forgetting mitigation literature [2]. A thorough analysis in the context of our work is however best deferred to a future paper.
>
> [1] Gu et al., 2024
>
> [2] https://arxiv.org/abs/1612.00796
>
> ---
>
> > **#4: More Fine-Grained Tasks Beyond Month Ordering**
>
> We appreciate your acknowledgment that the months-of-the-year task was intentionally chosen due to its clear interpretability and structured cyclic geometry. This choice enabled controlled experiments directly linking representation geometry and performance.
>
> However, we fully recognize your interest in finer-grained setups. We anticipate more realistic KE tasks—typically involving densely interconnected entities and facts—would exhibit even more severe representation shattering. Since greater interconnectedness implies more overlapping representations and deeper manifold entanglement (elaborated in our response to reviewer (see rVM5, #4), we expect edits in realistic scenarios to cause geometric distortions at least as severe as our cyclic-month edits. Our current findings represent a conservative estimate of the impact representation shattering can have in practical, large-scale knowledge editing scenarios; we plan to explicitly explore these richer domains as an immediate follow-up.
>
> ---
>
> > **#5: Generalization Beyond Cyclic Structures**
>
> We acknowledge your concern regarding our emphasis on cyclic structures, and fully agree that exploring other structures (e.g., trees or hierarchies) is crucial. Our current focus on cycles is deliberate: cycles are the simplest nontrivial graphs exhibiting global geometric constraints, ideal for precisely measuring and visualizing representation shattering. Their symmetric nature mitigates extraneous considerations like hierarchical relations, equivalence classes, and edge cases for root/leaf entities.
>
> As you note, our preliminary experiments on tree-structured KGs (App. G.3) align closely with our cyclic results: edits causing larger geometric displacement yield greater representation shattering (e.g., moving "Paris" to UK vs. Spain). Thus, we strongly believe the core phenomenon generalizes beyond cycles. Still, more systematic evaluations with alternative relational structures is needed—this is a key next step we intend to pursue in future work.
>
> ---
>
> > **#6: Code Release**
>
> We are committed to releasing our code for the camera-ready version for reproducibility (see rVM5, #6).
>
> ---
> ---
>
> **Summary:** We again thank the reviewer for their insightful feedback that has helped us emphasize the existing multi-step editing and layerwise analyses in our work, discuss catastrophic forgetting literature, and affirmed our intent to explore more complex setups. We hope our rebuttals address the reviewer's questions, and that they will continue to champion our paper's acceptance!

---

### Official Review · Reviewer_rub2 · 2025-03-11

**Overall Recommendation:** 2

**Summary:**

This paper explores the empirical finding that knowledge editing can degrade the general capabilities of large language models. The authors perform a synthetic in the setting of structured knowledge graphs (i.e. knowledge graphs with a cyclic or tree structure). They make the finding that the latent space activations of models adapt to the global structure of the knowledge graph. Next, they show empirically that both (a) corrective and (b) counterfactual edits shatter these representations. As a result, the post-editing performance of the model on both factual accuracy and reasoning capabilities becomes severely disrupted. The level of disruption is correlated with a notion of the "size" of the edit -- that is how far it diverges from the ground-truth knowledge graph. Finally, they provide evidence that this phenomenon also holds on a real Llama model on a task with a cyclic knowledge graph structure.

**Claims And Evidence:**

Claim 1) On synthetic structured knowledge graphs, the latent space encodes the geometry of the knowledge graph.

This is evidenced by low-dimensional visualizations of the latent space activations. In the main body of the paper, this is primarily shown on relatively simplistic cyclic structure. In the appendix, the authors additional provide some analysis of tree-like structures. While the cyclic structure does convincingly appear in their experiments, the tree like structure results are not convincing at all -- they only appear to show single examples and rely on very unclear and hard to discern arguments. It would be nice to have averaged R(D) measures for this setting just as was presented in the cyclic case. Hence, while they support their claim in a very limited setting of cyclical knowledge graphs, the general claim of explaining the failures of knowledge editing is not supported by this submission.


Claim 2) Knowledge editing techniques distort the geometry of the knowledge graph -- leading to degradation in the model. The extent to which this distortion occurs is captured by a "distance" of the edit (relative to the underlying KG).

This is primarily accomplished by way of qualitative analysis of feature visualization. Indeed, they do show that qualitatively, the visualization of the latent space changes structurally in a significant way. However, they do also quantify this using their measure of representation distortion. Thus, for the case of cyclical knowledge graphs, I believe the first claim is supported. The authors also observe that the amount of representation shattering correlates strongly with the extent of degradation of the model's capability. However, I do not believe that this actually establishes any "mechanistic" or causal claims about the change in representation geometry leading to worse performance which the authors claim is the case. This could be established better, for example, if the authors were able to show that the structure in the representation is **essential** for performing the evaluation tasks they consider. However, their current experiments do not rule out that there exists solutions with a different (or even disordered) latent space that could still perform the evaluation tasks.


Claim 3) Knowledge Shattering Occurs in Real LLM
The authors establish this in a pretrained Llama model on the representation of months. They show that months occupy a cyclic structure in the latent space and that applying ROME/MEMIT  to reassign different month names to different times in the year distorts this representation. Although this analysis is somewhat interesting, the task does not resemble the typical use-cases of knowledge editing (such as editing attributes of people/places/entities). In particular, this task appears to have been chosen due to its simple cyclical structure. For these reasons, this task in my opinion is insufficient to establish that representation shattering is actually the contributing phenomena for the "real world" tasks of knowledge editing.

**Essential References Not Discussed:**

N/A

**Experimental Designs Or Analyses:**

I have significant concerns about the experimental methodology used to evaluate ROME. The authors suggest in the supplementary material that they only try one choice of layer (the first layer) in their synthetic experiments because it is the only "early-site" layer. In general, I am somewhat confused about why this is the case and why the authors did not use the "causal tracing" mechanism introduced by the original ROME paper to select the location of the edit. In addition, the authors failed to list the number of layers used in the models for their synthetic setting--this also makes it hard to evaluate the validity of the experimental design.

**Methods And Evaluation Criteria:**

The evaluation criteria appears reasonable. As detailed below, I have questions about (a) why is ROME/MEMIT an appropriate method to use here and why was the layer for editing selected via the causal tracing mechanism (or tuned over).

**Other Comments Or Suggestions:**

This paper could also be made more stronger if the authors could demonstrate -- maybe in a very simplified theory setting-- why the geometry of the latent space is essential for solving the task. This would strengthen their causal claims about representation shattering being responsible (causally) for model degradation.

**Other Strengths And Weaknesses:**

At a high level, I have objections surrounding the underlying motivation for this work. As the authors specifically acknowledge in their appendix. ROME and related techniques are predicated on the functionality of MLP layers as implementing key-value storage via a linear association matrix. Prior  empirical works  [Geva, Mor, et al. "Transformer feed-forward layers are key-value memories." arXiv preprint arXiv:2012.14913 (2020); Geva, Mor, et al. "Dissecting recall of factual associations in auto-regressive language models." arXiv preprint arXiv:2304.14767 (2023).] have substantiated these hypotheses extensively (including the original ROME paper). The authors in this case propose a new task with global geometry of the latent space -- without verifying that it satisfies the underlying assumption of key-value storage-- and then show that ROME/MEMIT fail. This is unsurprising and it limits the applicability of this work to the many cases in which factual knowledge has actually been seen to obey the key-value storage assumption, but ROME/MEMIT still results in degradation. To put it another way, this paper appears to propose a setting (without adequate real-world justification of its validity) in which the assumptions of ROME/MEMIT are unsatisfied and then show that ROME/MEMIT fail. Can the authors comment a bit more about why we would expect ROME to be able t perform an edit in cases such as this where there is global structures?

**Questions For Authors:**

See above

**Relation To Broader Scientific Literature:**

This paper does related to an important collection of prior works regarding model degradations induced by model editing. The authors claim to provide a mechanistic hypothesis for these observations. However, the evaluation is not sufficiently robust to actually convince the reviewer that the mechanism they propose is actually the cause of the widely observed shortcomings of model editing in real-world settings.

**Theoretical Claims:**

Not applicable

---

> ### Author Rebuttal · Authors · 2025-03-31
>
> We thank the reviewer for appreciating our experiments and analysis of representation shattering. Please see our responses below.
>
> ---
>
> > **#1: Generality and Tree-Structured KGs**
>
> We acknowledge that our tree-based experiment in App. G.3 is preliminary, but believe further exploration of tree-shaped KGs is best reserved for follow-up work. Specifically, our paper aims to establish representation shattering, define evaluations to detect/visualize it, and demonstrate correspondence between synthetic and large-scale models. Cyclic graphs are uniquely suited for this: they are the simplest non-trivial graphs with global geometric constraints where post-edit effects can be exhaustively elicited. By contrast, trees lack this symmetry and introduce complexities around hierarchical relations, equivalence classes, and edge cases for root/leaf entities.
>
> We agree that tree-shaped KGs are an important next step, and we intend to pursue this direction. For this work, however, we prioritize simplicity—our cyclic results isolate the essence of representation shattering for the community to build on.
>
> ---
>
> > **#2: Mechanistically Causal vs. Correlational Evidence**
>
> While several of our experiments elicit correlational evidence, we emphasize our counterfactual editing experiments suggest a causal role for representation geometry in model capabilities (see Sec. 4.4/4.5, esp. Tab. 2, Fig. 6, Fig. 7). We summarize these results below.
>
> **Summary of counterfactual experiments.** We systematically vary counterfactual edit distance (CE distance) while holding all other variables fixed. For a given subject entity, edits with larger CE distances imply greater displacement in the representation manifold and higher $R(D_*)$. In reverse, higher $R(D_*)$ implies a larger CE distance. This manipulation approximates an intervention on representation geometry itself.
>
> Across all cases, higher CE distance leads to greater representation shattering and larger degradation in direct recall, logical inference, and compositional inference. While we cannot rule out models with scattered but functional representations, it is unclear how to obtain such a model to test this. We argue that the manipulation of CE distance—which directly impacts geometry—offers strong evidence for a causal link between shattering and performance loss.
>
> We will clarify that our experiments test for behaviors consistent with a mechanistic explanation, though we do not claim formal proof.
>
> ---
>
> > **#3: Realism of the "Months" Task**
>
> We agree the months-of-the-year task is simpler than real-world KE scenarios involving entity attributes. We selected it to isolate and quantify representation shattering in pretrained models.
>
> Though somewhat artificial, the fact that shattering arises in this clean cyclic setting suggests that richer domains—where knowledge is denser and more interdependent—would experience more severe effects. That is, if shattering already causes degradation in this minimal case, it likely plays a role in realistic tasks too. While we don't claim it explains all KE failures, our results show shattering is a major contributor. Exploring broader domains remains an important future direction (Sec. 5).
>
> ---
>
> > **#4: Layer Choice & Key-Value Assumption (ROME/MEMIT)**
>
> In all synthetic experiments, we use a 2-layer nanoGPT Transformer (explicitly stated in Sec. 3.3) Given this architecture, layer 1 is the only editing site compatible with the early-site assumptions of ROME. Further, causal tracing and MEMIT use a sliding window of layers: in a 2-layer model, any non-1 window size would encompass the whole model. We thus used layer 1 to avoid confusion, but early sanity checks with a singleton window re-implementation of causal tracing also confirmed layer 1 as the primary factual recall site. If the reviewer believes these results should be included, we can prepare them for the final version.
>
> We also qualitatively verified the key-value assumptions in exploratory stages. Prompts about the same entity yield clusters in the input space of the layer 1 MLP; prompts with different subjects resolving to the same object yield clusters in its output space. These patterns support the key-value framework underlying ROME/MEMIT as compatible with our synthetic setup. If it will help address reviewer's concerns, we are happy to add visualizations confirming these assumptions.
>
> ---
>
> > **#5: Code Release**
>
> We will release our code for the camera-ready version for reproducibility (see rVM5, #6).
>
> ---
> ---
>
> **Summary:** We again thank the reviewer for their thoughtful feedback that helped us clarify the rationale behind our synthetic design, show how our experiments suggest a causal link between representation geometry and model performance, acknowledge the simplicity of our Llama task as deliberate, and justify our editing layer choices and key-value assumptions. We hope our rebuttals address the reviewer's questions and that they will champion our paper's acceptance!

---

### Official Review · Reviewer_rVM5 · 2025-03-14

**Overall Recommendation:** 3

**Summary:**

This paper explores the impact of Knowledge Editing (KE) on Transformer models and introduces the concept of Representation Shattering. The authors argue that modifying specific facts in the model will destroy its broader internal knowledge structure, leading to reduced fact recall and reasoning capabilities. The authors design a synthetic task using structured knowledge graphs to systematically analyze this effect and validate it on pre-trained LLMs (Llama and Mamba), showing that KE leads to widespread distortions in model representations. This paper provides some empirical and theoretical insights into the potential mechanisms of knowledge editing failures.

**Claims And Evidence:**

1. Knowledge Editing harms the overall capability of the model, not just the modified facts.
- The author tested on direct recall, logical reasoning, and combinatorial reasoning tasks and found that the accuracy of the model dropped significantly after KE.
2. Claim: Representation fragmentation is the core mechanism that causes model knowledge degradation.
- The paper measures the change in model representation before and after KE by the Frobenius norm difference and finds that it is strongly correlated with the decline in model performance.
3. Claim: The larger the edit distance, the more severe the representation fragmentation.
- Evidence: On the synthesis task and LLM task, the study found that modifying facts with a longer distance (such as "January → February" vs. "January → June") is more damaging to representation.
4. Limitations: The paper mainly verifies this phenomenon from an experimental perspective, but the theoretical explanation of why the internal mechanism of the Transformer is so fragile still needs to be further improved.

**Essential References Not Discussed:**

WISE: Rethinking the Knowledge Memory for Lifelong Model Editing (https://arxiv.org/abs/2405.14768)

This paper proposes the WISE algorithm, which achieves an effective bridge between long-term memory and working memory in large language models by designing a dual-parameter memory scheme and a knowledge sharding mechanism to solve the impossible triangle problem of reliability, generalization and locality in lifelong model editing.

**Experimental Designs Or Analyses:**

Using Structured Knowledge Graphs provides a highly controllable KE research environment. The results are verified on the pre-trained Llama and Mamba, which improves the external validity of the research. More Transformer variants (such as Bert, GPT-4, Mixtral) need to be tested to enhance the generalizability of the conclusions. The paper does not deeply analyze how different Transformer components (such as MLP vs. attention head) are affected by KE. It is recommended to add more micro-level experiments.

**Methods And Evaluation Criteria:**

The authors use Structured Knowledge Graphs as a Toy Setting for Investigating the Impact of KE.
The evaluation includes three key metrics: direct recall, logical inference, and compositional inference.
The study did not consider real tasks (complex logic), which limits the generalization ability to real-world applications.

**Other Comments Or Suggestions:**

Some suggestions: Extend theoretical analysis to explain why Transformer representations are so susceptible to KE. Test the impact of KE on retrieval enhancement models (such as RAG, RETRO) to explore whether external memory can alleviate representation fragmentation.
Compare with the latest KE method (WISE) to verify the applicability of the research conclusions.

**Other Strengths And Weaknesses:**

Strengths:
1. Designed novel synthetic tasks that can precisely control the KE effect.
2. Provided clear empirical analysis and proposed methods to quantify representation fragmentation.
3. It has important implications for the design of future KE methods.

Weaknesses:
1. Mainly based on small-scale synthetic tasks, the applicability to real-world applications still needs further verification.
2. It did not explore the underlying mechanism of Transformer representation construction in depth, and the theoretical explanation can still be improved.
3. The experiments are all based on pre-trained models, and fine-tuning can be considered; the paper does not provide a detailed ablation study to isolate the impact of pretraining versus fine-tuning.

**Questions For Authors:**

1. Is there a difference in representation fragmentation between different layers? For example, does KE have the same effect on MLP layers vs. attention layers?
2. The cyclic knowledge in the article topic is not common in the real world. How does it perform when faced with complex tree-structured knowledge?
3. Can you provide UMAP/t-SNE visualizations before and after KE? Intuitively show how representation fragmentation occurs.
4. Can retrieval enhancement architectures (such as RAG, RETRO) reduce representation fragmentation?

**Relation To Broader Scientific Literature:**

The research results are related to issues such as AI reliability and knowledge plasticity and have made certain contributions to AI security research.

**Theoretical Claims:**

The study shows that the model's knowledge representation is structured rather than independent facts stored in isolation, and verifies that the larger the edit distance, the more severe the representation fragmentation, and provides experimental support. The paper lacks an explanation for why the Transformer representation is so susceptible to KE.

---

> ### Author Rebuttal · Authors · 2025-03-31
>
> We thank the reviewer for appreciating our experiments and analysis of representation shattering. Please see our responses below.
>
> ---
>
> > **#1: Scope of Models and Methods Tested**
>
> We emphasize our evaluation already spans multiple architectures and scales: small Transformer models trained from scratch on synthetic knowledge graphs, pretrained decoder-only models without fine-tuning (GPT-2), pretrained decoder-only models with instruction fine-tuning (Llama 3 8B Instruct), and pretrained structured state-space models without fine-tuning (Mamba 2.8B). All consistently exhibit representation shattering that scales with counterfactual edit distance (see Sec. 4.5 and App. G.1).
>
> Regarding the reviewer's suggested settings, we note the following.
>
> - **BERT**: Modern KE methods exclusively focus on causal decoder-only models (ROME, MEMIT, PMET, AlphaEdit), due to which bidirectional encoder models like BERT are not amenable to analysis under the purview of our work.
> - **GPT-4**: We note GPT4 is a closed-source model. This precludes parameter-space editing and replicating our methods.
> - **Mixtral**: We note our Llama 3 8B experiments already feature similar scale, decoder-only construction, and instruction tuning as a Mixtral model. While MoE architecture effects are interesting, there is no precedent for such an experiment, so this lies outside our scope for now.
>
> - **RAG and WISE**: RAG, RETRO, and KE protocols like WISE operate under different assumptions (e.g., inference-time components). Thus, they extend beyond our current scope of analyzing direct parameter-editing methods.
>
> Nevertheless, we agree the directions above are promising. We'll include these in an expanded "Future Directions" section, citing WISE.
>
> ---
>
> > **#2: Layerwise Analysis of Representation Shattering**
>
> Current KE methods use causal tracing to identify the layer to edit, often focusing on early MLP layers. Modifying this step would require a new KE protocol, which is beyond our scope. Nevertheless, our manuscript includes visualizations from different sub-layers and depths (see App. F.3 and G.2) that show the spatial extent of shattering. We are open to extending these analyses if helpful (e.g., as outlined in response to reviewer kjUD, #2).
>
> ---
>
> > **#3: Visualization Methods**
>
> We refer the reviewer to our comment justifying the use of Isomap in response to reviewer gDBJ, #3. In brief, we use Isomap due to its ability to preserve geodesic distances, hence observing how cyclical manifolds distort under KE. UMAP/t-SNE show clustering, but Isomap's focus on global geometry better captures topological collapses from KE.
>
> ---
>
> > **#4: Theoretical Explanation for Transformer Fragility under KE**
>
> Though mainly empirical, our study aligns with a broader theoretical framework.
>
> - Transformers store factual associations in key-value pairs within MLP layers [1, 2] corroborated by our synthetic model analysis (see also response to rub2, #4).
> - Parameter sharing and superposition cluster unrelated facts in overlapping subspaces [1], making them vulnerable to unintentional interference.
> - Entities and relations often form structured manifolds (e.g., cycles, hierarchies), which aid compositional inference [4].
> - KE methods (ROME, MEMIT, etc.) enact local weight updates that deform these manifolds, causing representation shattering for unedited facts residing in shared sub-regions.
>
> Hence, we believe fragility arises from the entangled, compressed nature of factual storage, rather than from simply retaining knowledge. We will include this discussion in the final version of the paper.
>
> [1] Geva et al., 2020
>
> [2] Meng et al., 2022a
>
> [3] https://transformer-circuits.pub/2023/toy-double-descent/index.html
>
> [4] Engels et al., 2024
>
> ---
>
> > **#5: Evaluation on Tree-Structured Knowledge**
>
> App. G.3 explores KE on a tree-structured city–country graph using GPT-2 XL. While trees are more challenging to visualize than cyclical manifolds, smaller-distance edits (e.g., relocating "Paris: to "Spain:) exhibit less shattering than those with larger distance (e.g., "UK:). This supports our hypothesis that shattering severity scales with the manifold distance of the edit.
>
> ---
>
> > **#6: Supplemental Materials and Code Release**
>
> See Apps. A/B/C for documentation on pseudocode, hyperparameters, data, and architectures; we are happy to provide more!
>
> We also highlight App. A.1, where we state our commitment to publicly releasing the source code for our experiments—covering both synthetic and naturalistic settings—in the camera-ready version of this paper. This ensures transparency and reproducibility.
>
> ---
> ---
>
> **Summary:** We again thank the reviewer for their insightful comments that have helped us clarify our design choices, the constraints of current KE layer selection methods, our use of Isomap, and the theoretical underpinnings of Transformer fragility. We hope our rebuttals address the reviewer's questions and that they will champion our paper's acceptance!

---

### Official Review · Reviewer_gDBJ · 2025-03-14

**Overall Recommendation:** 4

**Summary:**

This paper proposes a fundamental principle to understand why existing Knowledge Editing (KEs) methods often introduce unexpected cascading effects on knowledge not tampered with during edition and cause the edited LLMs to yield inconsistent reasoning results. Specifically, the authors argue that, especially for relational knowledge like those represented by Knowledge Graphs (KG), the representations of the entities in a good pre-trained model will demonstrate geometries that are consistent with the underlying KG. However, KE procedures would invariably break or shatter this representation geometry, thus causing troubles in downstream reasoning tasks. To support this claim, the authors performed extensive empirical studies, both quantitative and qualitative, on multiple SoTA KE methods (ROME, MEMIT, PMET, & AlphaEdit), multiple LMs (2-layer transformer and pretrained LLMs like Llama3 8B, Memba), on both cyclic-structured and tree-structured knowledge graphs in a synthetic setting.

**Claims And Evidence:**

Yes, the claims made in the paper are supported by clear, convincing, comprehensive empirical evidence, both with quantitative measurements (vai representation distance metric $R(D)$) and qualitative visualization (via Isomap of model's internal representation).

It is noteworthy that the authors perform the study using multiple KE methods, with both pertaining a small LM and using already-pretrained LLMs, on both cycle-structured and tree-structured KGs. The comprehensiveness and extensiveness of the empirical effort make me confident that the representation-shattering phenomenon should be quite universal and general.

**Essential References Not Discussed:**

As far as I know, there is no other essential references not discussed. The paper's reference list is quite comprehensive.

**Experimental Designs Or Analyses:**

I checked the soundness/validity of the experimental designs and analyses. Overall, I believe the methodology makes sense. And it is noteworthy that the authors have done an arguably very extensive and comprehensive empirical study, covering multiple settings, models, KE methods, and knowledge structures.

However, I have a question regarding the evaluation task and would appreciate the author's clarification:

Q3: I don't fully understand why would the logical inference task in evaluation make sense. The authors describe this task as follows:

> Logical inference accuracy measures the accuracy on a subset of held out relations that can be inferred from other relations

Is it correct that these held-out relation tokens were never seen by the model during pre-training stage via next-token prediction? If so, wouldn't it mean that these held-out relation tokens are completely unseen by the model at inference time? If so, how does it make sense that we can expect the model to know what these held-out relation tokens mean, and to give correct answers on prompts involving these held-out relation tokens?

**Methods And Evaluation Criteria:**

Overall I believe the proposed investigative method makes sense. However, I do have several concerns/questions regarding the evaluation metric.

Q1: To quantify the extent of representation shattering, the authors proposed a metric measuring "representation distortion", which is defined as
$$R(D_*) = \frac{\vert\vert D_* - D_{\emptyset} \vert\vert_F}{ \vert D_{\emptyset} \vert_F  },$$
where $D_{\emptyset}$ and $D_*$ are the pairwise Euclidean distance matrix of the entities's vector representations in the original and unedited models respective.

However, I don't fully understand what should we expect from this metric? Specifically, suppose there exists a perfectly edited model that somehow has its entities' internal representations' geometry maintained and avoids shattering. Shall we expect that this model have $R(D_*) = 0$? I guess the answer to this question is no. Take the edits shown in Figure 3 as an example. Suppose we are doing the following two counterfactual edits:

1.I_C2 = 3→2,
1.I_C3 = 2→3.

That is, suppose the counterfactual edits intend to exchange entity 2 with entity 3, or "rename" entity 2 to 3 and entity 3 to 2. Then, our perfect edited model still maintains the representation structure by simply exchange the representation of entity 2 with entity 3.

In this case, $D_*$ and $D_{\emptyset}$ are different matrices, even though the set of all the entries (all the pairwise Elucidean distances) in these two matrices is the same. More precisely, $D_*$ and $D_{\emptyset}$ are isomorphic up to permutation. However, **because R(D) does not account for permutations and it is not invariant to permutations**, $R(D_*)$ is not 0 for this perfect edited model. Furthermore, the specific value of $R(D)$ for this perfect model also depends on which pair of entities I exchanged via counterfactual edits, because different pairs of entity exchange will give difference matrix, $D_* - D_{\emptyset}$.

Hence, I believe the $R(D)$ metric might be inconsistent in that it does not give a 0 value for the perfect model, and its specific value might be sensitive to the identity of entities involved in knowledge edits. If the $R(D)$ metric can be somehow made to be invariance to permutation to entity identities, I belive this issue could be resolved.

Q2: Regarding the qualitative studies, the authors adopt the Isomap method. Is there a particular reason why Isomap is necessary, or the best dimensionality-reduction approach suitable for this task? For instance, is there a reason why other methods, such as simple PCA, wouldn't work or wound't be applicable?

**Other Comments Or Suggestions:**

I have several other comments on readability and typos:

- S1: (Page 5) Readability of the $R(D)$ metric definition: it is not immediately clear what is the shape of these distance matrices $D$, and that they are defined over all entity tokens (actually, is it true? Does it also account for relation tokens?)  It would be great to explicitly say something like "$D_{ij}$ is a scalar value of the Euclidean distance between entity i and j's representations."

- S2: (Page 5) Description of Evaluation (unseen facts): I feel that the description for this part is too vague, unclear and confusing. In particular, I did not understand what was the prompts for the compositional inference task and the logical inference task, and how are they different from each other. The authors did well in the previous paragraph (Evaluation (seen facts)). It would be great if the authors can also show, precisely, what does the input prompt for the compositional inference task and the logical inference task look like.

- S3: I checked Appendix G.3 and the findings on the tree-structured KG is quite interesting. It would be great if the authors can mention these findings in one or two sentences in the main paper. I think this is quite important particularly because tree-structure KG is a more natural structure than cyclic KGs and more relevant to real world.

 - S4: some typos:
    - (Page 3) Definition 3.1: it is written $f = (x_i, r, x_j) \in R$ but R is the relation set. It should be $f = (x_i, r, x_j) \in F$ because F is the fact set.
    - (Page 4) 3.3 Experimental Setup - Data generation process $x_i r_i x_{i+1}$ should be $x_i \vec{r_i} x_{i+1}$.

**Other Strengths And Weaknesses:**

Apart from the strengths mentioned in the previous sections, it's also commendable that the authors did quite a decent job in explaining how the dataset is constructed, in particular, what are the cyclic orders and what is the structure of the synthetic KG (Fig 3). This is quite complicated a conceptual construct but the exposition is easy to follow.

**Questions For Authors:**

Please check my questions Q1, Q2, Q3 above. In addition, I have one other question:

Q4: My takeaway from this paper is that knowledge edits, in particular counterfactual ones, shatter representation because they introduce inconsistencies.

What if the "user" who makes these edits is more "careful" and makes a set of counterfactual edits that ensures consistency? For instance, say the "user" intends to exchange the name of two entities, and provides a set of comprehensive counterfactual edits where each affected relational triplet is included. With this careful "user", will the model's internal representation still be shattered? In other words, can we attribute the representation shattering to incomprehensive and/or inconsistent counterfactual edit queries, or is it that there is something else that is fundamentally wrong, that the internal representation of LLMs will invariably shatter even if the "user" is sufficiently careful and the edits are sufficiently comprehensive?

---

Overall, despite some concerns on the validity of the evaluation metric, I believe that the comprehensiveness of the empirical results shows that representation shattering should be quite a universal phenomenon and that this paper is an impactful contribution to the field. Thus, I recommend this paper for acceptance.

**Relation To Broader Scientific Literature:**

This paper proposes a fundamental and principled understanding as to why KE methods in general introduce unwanted cascading effects in the model's internal knowledge and hence break the model's reasoning performance.

I believe this is a very important contribution to the field, and will provide a guidance to future work on how to address these fundamental issue of KE.

**Theoretical Claims:**

This paper does not have theoretical components that require proofs.

---

> ### Author Rebuttal · Authors · 2025-03-31
>
> We thank the reviewer for appreciating our experiments and analysis of representation shattering. Please see our responses below.
>
> ---
> > **#1: $R(D_*)$ and Permutation Invariance**
>
> Great question! We intentionally made our distortion metric $R(D_*)$ to not be permutation-invariant, i.e., $R(D_*) > 0$ even if $D_*$ is a permutation of $D_{\varnothing}$. This is because $R(D_*)$ tracks *how much each entity's position in the representation space, as identified by its token, has changed*—not whether the manifold is isomorphic under another labeling.
>
> For example, in your suggested swap of entity 2 with entity 3, the manifold may remain consistent, but the label-respecting configuration changes, and so do the model's semantics. A zero value would suggest no parameter shift even if outputs differ.
>
> A permutation-invariant version (e.g., via minimum matching) would conflate isomorphic configurations and assign zero cost to degenerate cases—like total entity permutation—despite incoherence. For our purposes, permutation invariance is a feature. We will clarify this in the final version of the paper.
>
> ---
> > **#2: Shape of $R(D_*)$**
>
> $D$ is an $n \times n$ matrix over $n$ entities, where $D_{ij}$ is the Euclidean distance between representations of $x_i$ and $x_j$. We exclude relation tokens. We'll state this in the final version.
>
> ---
> > **#3: Isomap vs. PCA**
>
> Isomap preserves geodesic distances along the manifold, which is essential for visualizing non-linear geometries (e.g., rings, torii) [1,2]. Meanwhile, PCA, being linear, can collapse or stretch such structures, obscuring effects like representation shattering. This makes us prefer Isomap for our analysis, since faithful projections of the original topology turn out to be critical for interpreting how KE affects global geometry. We use PCA as well when linearity suffices (see App. F.2).
>
> [1] https://www.nature.com/articles/s41593-019-0460-x
>
> [2] https://www.nature.com/articles/s41583-022-00642-0
>
> ---
> > **#4: Held-Out Tokens and Logical Inference**
>
> We note that "held-out" refers to specific facts, not entire relation tokens. Specifically, when generating the training dataset, we drop sequences that state one direction of a pair of conjugate facts with fixed probability $p$. That is, for any entity $x_i$ and relations $r, r'$, suppose the fact $(x_i, r, x_j)$ always implies $(x_j, r', x_i)$ (i.e. $r=$`I_C1` and $r'=$`I_A1`). Our DGP may drop *one of* these *facts* from the training data (with probability $p$). Even if $(x_j, r', x_i)$ is absent, one can still learn the conjugacy of $r$ and $r'$ and the existence of $x_i$ and $x_j$ via other examples; however, failure to infer this relation indicates the model has rote memorized relations rather than understanding the global structure.
>
> This discussion is detailed in App. B, and we'll clarify this further in the final version.
>
> ---
> > **#5: Bulk Edits and Avoiding Shattering**
>
> In theory, we agree that a fully self-consistent batch of edits could reduce shattering. In practice, we observe:
>
> - KE methods like ROME and MEMIT maximize the likelihood of edited facts without explicit constraints on representation geometry. Like task-specific fine-tuning, this can induce interference and catastrophic forgetting.
>
> - Batch sizes are small because of compute limits. Edits are not independent or additive, so parallel updates can introduce inconsistencies, with degradation compounding over edits [1].
>
> - Enumerating all semantically connected facts is intractable. Editing "Eiffel Tower is in ~~Paris~~ Rome" would require accounting for every related event or entity; a self-consistent closure is undefined.
>
> Thus, while comprehensive edits may reduce shattering, they do not eliminate it.
>
> [1] Yoon et al., 2024
>
> ---
> > **#6: Prompt Format for Inference**
>
> - **Compositional**: The model is given a chain of relations (e.g., $x_i\, r_1\, r_2$) and must produce $x_k$, with $(x_i, r_1, x_j)$ and $(x_j, r_2, x_k)$ seen separately in training but not composed. Input: $\text{ctx}\, x_i\, \vec{r}$ where $\vec{r} = r_1r_2$.
>
> - **Logical**: The model is asked about a fact $(x_j, r', x_i)$ where $(x_i, r, x_j)$ was seen but $r'$ was withheld. If the model understands relation symmetry, it can infer the inverse.
>
> We'll further clarify these formats in the final version.
>
> ---
> > **#7: Tree-Structured KG (App. G.3)**
>
> Thank you for the suggestion to add a forward reference—we will do so in Sec. 4.5.
>
> ---
> > **#8: Code Release**
>
> We will release our code for the camera-ready version (see response to rVM5, #6).
>
> ---
> > **#9: Typos**
>
> Thank you! We'll fix these typos in the final version.
>
> ---
> ---
> **Summary:** We again thank the reviewer for their detailed review, which helped emphasize the design of our distortion metric $R(D_*)$, justify our visualization strategy, address inference task design, and explain the limits of bulk editing. We hope our rebuttals address the reviewer's questions and that they will champion our paper's acceptance!

---

### Decision · Program_Chairs · 2025-05-01

**Decision:**

Accept (poster)

**Comment:**

The authors develop a hypothesis (and evidence supporting it) regarding why cascading errors ensue after knowledge editing (KE) in LLMs. To accomplish this, they introduce the concept of representation shattering, the case where modifying the representation of the targeted entity distorts the representation of other related entities, thus distorting the relationships between entities and thus the ability to infer unseen (explicit) information between entities within this relation network. They specifically examine the case of structured information in knowledge graphs and show that 'good' pre-trained models with exhibit geometric structures that are consistent with the underlying KG structure. However, KE methods even when applied to a small number of these entities 'shatter' this geometric structure and thus introduce cascading errors. Empirical validation is performed with multiple widely-studied KE methods (e.g., ROME, MEMIT, PMET, AlphaEdit) with multiple (L)LMs (e.g., a 2-layer transformer network, Llama3-8B, Mamba) based on cyclic-structured and tree-structured knowledge graph content (in synthetic experiments).

The strengths of this submission identified by the reviewers include:
- This paper builds on recent explorations for LLMs based on synthetic data generation processes (more widely used for interpretability studies) to provide some theoretical insight into why knowledge editing hasn't proven successful at scale, despite being intuitively appealing. While KE hasn't proven successful, this is the first work any of us are aware of that provides an explanation.
- The study includes multiple KE methods as applied to  a simple transformer LM and existing pre-trained LLMs on multiple KG structures. The visualizations and quantitative results are consistent across experiments, providing solid evidence for 'representation shattering' being a strong hypothesis.
- The experimental procedures are well explained and the paper is well-motivated and well-written overall.

Conversely, limitations of this work identified by the reviewers include:
- This work doesn't provide solutions for mitigating representation shattering (even in discussion regarding future directions). Do we expect solutions to be via the KE protocol or in the transformer architecture (or some combination).
- While a non-negligible step in understanding, this is still limited theory that is well-supported empirically, but there isn't strong analytical analysis. As reviewer rub2 points out, it isn't shown that the structure in the representation is essential (as it isn't identifiable) as scattered representations may still be functional (and performant). However, this was well-discussed during rebuttal.
- The experiments are for very controlled settings and there isn't even discussion regarding how this relates to more complex reasoning tasks (i.e., are the constructed settings illustrative, representative, or adversarial?). This was also well-discussed during rebuttal.

Overall, I think this is a promising work that will likely inspire further work, potentially across multiple LLM characteristics. Using synthetic data generation procedures to examine the dynamics and relationships between representation and generation is relatively new and this study was well-constructed. Assuming that knowledge editing remains as a potential solution to information/knowledge updating, the representation shattering formalism may point a way to mitigating such procedures (e.g., with some sort of representation regularization, data mixing, etc.). Additionally, such findings could lead to a general understanding of 'updating LLMs' or training procedures in general. Thus, while the findings could be interpreted as preliminary (i.e., not actionable), I think the community would find the results and procedure interesting and potentially insipring.